# Lissencephaly-1 is a context-dependent regulator of the human dynein complex

Janina Baumbach[1], Andal Murthy[1,2†], Mark A McClintock[1†], Carly I Dix[1], Ruta Zalyte[2], Ha Thi Hoang[1], Simon L Bullock[1*]

[1]Division of Cell Biology, MRC Laboratory of Molecular Biology, Cambridge, United Kingdom; [2]Division of Structural Studies, MRC Laboratory of Molecular Biology, Cambridge, United Kingdom

**Abstract** The cytoplasmic dynein-1 (dynein) motor plays a central role in microtubule organisation and cargo transport. These functions are spatially regulated by association of dynein and its accessory complex dynactin with dynamic microtubule plus ends. Here, we elucidate in vitro the roles of dynactin, end-binding protein-1 (EB1) and Lissencephaly-1 (LIS1) in the interaction of end tracking and minus end-directed human dynein complexes with these sites. LIS1 promotes dynactin-dependent tracking of dynein on both growing and shrinking plus ends. LIS1 also increases the frequency and velocity of processive dynein movements that are activated by complex formation with dynactin and a cargo adaptor. This stimulatory effect of LIS1 contrasts sharply with its documented ability to inhibit the activity of isolated dyneins. Collectively, our findings shed light on how mammalian dynein complexes associate with dynamic microtubules and help clarify how LIS1 promotes the plus-end localisation and cargo transport functions of dynein in vivo.

*For correspondence: sbullock@mrc-lmb.cam.ac.uk

†These authors contributed equally to this work

Competing interests: The authors declare that no competing interests exist.

## Introduction

Cytoplasmic dynein-1 (hereafter referred to as dynein) is a 1.4 MDa, multi-subunit motor complex that is responsible for the vast majority of cargo transport towards the minus ends of microtubules in eukaryotic cells. Dynein's cargoes include organelles, vesicles, aggregated proteins, mRNAs and pathogenic viruses (*Carter et al., 2016*; *Dodding and Way, 2011*; *Hoogenraad and Akhmanova, 2016*). In addition to its function on motile cargoes, dynein plays important roles in organising microtubules in mitotic and interphase cells (e.g. *Bader and Vaughan, 2010*; *Dujardin et al., 2003*; *Dujardin and Vallee, 2002*; *Koonce et al., 1999*; *Laan et al., 2012*; *Nguyen-Ngoc et al., 2007*; *Ten Hoopen et al., 2012*; *Varma et al., 2008*).

Dynein – like all cytoskeletal motors – functions within cells on a dynamic network of tracks. Whereas the microtubule minus ends are often stabilised in vivo, the plus ends alternate between phases of polymerisation ('growth') and depolymerisation ('shrinkage')—a process known as dynamic instability. Microtubule plus ends are stabilised during growth by a cap of tubulin-GTP, with depolymerisation triggered when the GTP bound to $\beta$-tubulin at the microtubule tip is hydrolysed (*Carlier, 1982*; *Mitchison and Kirschner, 1984*). The dynamic nature of microtubules facilitates exploration of the cytoplasm in order to establish and remodel contacts with cellular components (*Kirschner and Mitchison, 1986*; *Lomakin et al., 2009*; *Mimori-Kiyosue and Tsukita, 2003*; *Tamura and Draviam, 2012*).

Dynein is detected along the lattice of the microtubule, but is enriched at the plus end in a wide variety of eukaryotic cells (*Han et al., 2001*; *Kobayashi and Murayama, 2009*; *Lee et al., 2003*; *Lenz et al., 2006*; *Ma and Chisholm, 2002*; *Moughamian et al., 2013*; *Schuster et al., 2011*; *Sheeman et al., 2003*; *Splinter et al., 2012*; *Vaughan et al., 1999*). Association of the motor

complex with the plus end is important for its microtubule organising functions at the cell cortex (**Lee et al., 2003**; **Markus and Lee, 2011a**, **2011b**; **Sheeman et al., 2003**) and for efficient initiation of retrograde cargo transport in polarised cells, including neurons (**Lenz et al., 2006**; **Lloyd et al., 2012**; **Moughamian and Holzbaur, 2012**; **Moughamian et al., 2013**; **Zhang et al., 2010**).

Cellular and biochemical experiments have provided insights into how dynein is recruited to microtubule plus ends in mammalian cells. A key player in this process is p150$^{Glued}$ (hereafter p150), a subunit of the dynactin complex. Dynactin is a 1.1 MDa complex of 11 distinct components that co-precipitates with dynein and is critical for its plus-end targeting and cargo transport functions in vivo (**Moughamian et al., 2013**; **Schroer, 2004**; **Splinter et al., 2012**). p150 contains a CAP-Gly domain that can bind directly to members of the end-binding (EB) protein family such as EB1 or EB3 (**Bjelić et al., 2012**; **Honnappa et al., 2006**). EB proteins dynamically track polymerising plus ends through repeated transient binding events on a structure associated with the GTP- or GDP.Pi-tubulin conformation (**Akhmanova and Steinmetz, 2015**; **Bieling et al., 2008**, **2007**; **Maurer et al., 2011**, **2012**; **Zanic et al., 2009**; **Zhang et al., 2015**). p150 can also contact the intermediate chain of the dynein complex (**Karki and Holzbaur, 1995**; **King et al., 2003**; **Vaughan and Vallee, 1995**). These data suggest a mechanism for linking dynein to growing plus ends in which p150 acts as a bridge between EB proteins and the motor complex. Consistent with this model, a truncated version of a tissue-specific p150 isoform induces tracking of the human dynein complex on EB1-associated plus ends of growing microtubules in vitro (**Duellberg et al., 2014**). However, these experiments did not address how dynein associates with the plus ends of microtubules in the presence of the full dynactin complex.

It is also not clear how dynein complexes that undergo processive, minus end-directed motion are recruited to dynamic microtubules in mammalian cells. Reconstituting this process in vitro has been very challenging because individual mammalian dynein complexes rarely exhibit processive motion (**McKenney et al., 2014**; **Miura et al., 2010**; **Schlager et al., 2014**; **Trokter et al., 2012**). However, a way of activating minus end-directed motion of single mammalian dynein complexes in vitro has been recently described (**McKenney et al., 2014**; **Olenick et al., 2016**; **Schlager et al., 2014**; **Schroeder and Vale, 2016**). Association of dynein with both the dynactin complex and a cargo adaptor results in frequent processive movements along microtubules. In the context of a dynein-dynactin-cargo adaptor complex, the motor also translocates faster and has a greater force output (**Belyy et al., 2016**; **McKenney et al., 2014**). Activation of dynein by dynactin and a cargo adaptor presumably allows robust minus end-directed motion to be coupled to the availability of cargo. The activating cargo adaptor that is best characterised is Bicaudal-D2 (BICD2), a coiled-coil protein that links dynein to several cargoes, including Golgi-derived vesicles and nuclear pore complexes (**Hoogenraad and Akhmanova, 2016**; **Hoogenraad et al., 2003**; **Splinter et al., 2012**). BICD2 uses its N-terminal regions (BICD2N) to associate with dynein and dynactin and its C-terminal regions to recruit cargoes (**Hoogenraad and Akhmanova, 2016**).

The Lissencephaly-1 (LIS1) protein is another key player in the regulation of dynein function. Reduced expression of LIS1 causes the neurodevelopmental disease type 1 lissencephaly in humans by perturbing neuronal proliferation and migration (**Moon and Wynshaw-Boris, 2013**). LIS1 forms a homodimer (**Kim et al., 2004**) that interacts directly with the dynein complex (**Huang et al., 2012**; **McKenney et al., 2010**; **Sasaki et al., 2000**; **Tai et al., 2002**; **Toropova et al., 2014**) and is essential for its enrichment at dynamic microtubule plus ends in mammalian cells (**Splinter et al., 2012**). LIS1 is not, however, needed for association of dynactin with plus ends (**Splinter et al., 2012**), suggesting a specific role in recruitment of dynein to these sites. LIS1's role in plus-end tracking of mammalian dynein has also not been addressed in previous in vitro reconstitution assays.

In addition to regulating association of dynein with microtubule plus ends, LIS1 is required for efficient transport of many dynein-associated cargoes (**Dix et al., 2013**; **Egan et al., 2012**; **Lam et al., 2010**; **Lenz et al., 2006**; **Pandey and Smith, 2011**; **Reddy et al., 2016**; **Splinter et al., 2012**; **Yi et al, 2011**; **Zhang et al., 2003**). In vitro experiments have shown that LIS1 increases the affinity of the isolated motor complex for microtubules and suppresses its mechanochemical activity (**Huang et al., 2012**; **McKenney et al., 2010**; **Torisawa et al., 2014**; **Toropova et al., 2014**; **Yamada et al., 2013**, **2008**). The findings that LIS1 inhibits the activity of isolated dyneins in vitro yet promotes cargo transport by the motor in vivo appear somewhat contradictory. To reconcile these observations, it was proposed that LIS1 improves ensemble function of dynein on high-load cargoes by allowing the motor to enter into a persistent force producing state (**McKenney et al.,**

*2010*; *Yi et al, 2011*). This model cannot, however, readily account for the observation that LIS1 also promotes the transport of small cargoes in at least some cell types (*Dix et al., 2013*; *Egan et al., 2012*). An alternative model is that LIS1 promotes cargo transport by targeting dynein complexes to the microtubule, with transport initiation triggered by dissociation of LIS1 from the cargo-motor assembly (*Egan et al., 2012*; *Lammers and Markus, 2015*; *Markus et al., 2011*; *Yamada et al., 2013*). This model has predominantly received support from elegant imaging and genetic studies in fungal systems, and it is not clear if LIS1 functions equivalently in mammalian cells. These models for LIS1 regulation of dynein activity are, of course, not mutually exclusive. Moreover, LIS1 could have additional roles in regulating dynein transport in vivo that were not evident in the in vitro reconstitution assays performed to date.

Here, we use total internal reflection fluorescence (TIRF)-microscopy to investigate in vitro the roles of EB1 and LIS1 in plus-end tracking of human dynein in the presence of the full dynactin complex. We also investigate the influence of these proteins on interactions of minus end-directed dynein complexes with dynamic microtubules by including dynactin and BICD2N in the assay system. Our approach elucidates how mammalian dynein function is spatially regulated on dynamic microtubules and reveals context-dependent effects of LIS1 that help clarify its in vivo functions.

## Results

### LIS1 stimulates plus-end tracking of dynein complexes on growing microtubules in the presence of EB1 and dynactin

We developed assay conditions that are permissive for the interaction of non-processive and minus end-directed mammalian dynein complexes with microtubules exhibiting dynamic instability (*Figure 1*; Materials and methods). Microtubules were polymerised from fluorescent tubulin in an imaging chamber in the presence of GTP. Polymerisation occurred from pre-assembled seeds that had been stabilised with the non-hydrolysable GTP analogue GMPCPP and adhered to the glass surface using a streptavidin-biotin linkage (*Bieling et al., 2007*; *Duellberg et al., 2014*). The full human dynein complex was produced in insect cells and fluorescently labelled using a SNAP tag at the

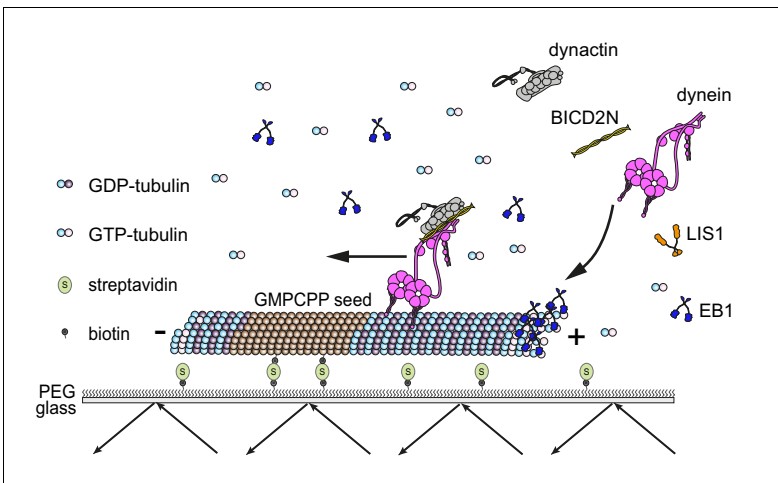

**Figure 1.** Diagram of the in vitro assay system. Dynamic microtubules are produced from stable, immobilised seeds and the behaviour of human recombinant dynein complexes studied in the presence and absence of the indicated regulatory co-factors, including activators of processive, minus end-directed motion (straight arrow). − and + refer to the microtubule minus and plus end, respectively. Purified dynein and its regulators were incubated together before diluting in a solution containing EB1 and free tubulin, and injection into flow chambers that had stable microtubule seeds preadsorbed on the glass.

The following figure supplement is available for figure 1:

**Figure supplement 1.** Purity of protein preparations.

N-terminus of the heavy chain subunit ([*Schlager et al., 2014*]; see *Figure 1—figure supplement 1* for data on the purity of this and other protein preparations used in the study). Dynein complexes were then introduced into the imaging chamber in the presence of a saturating concentration of ATP. The concentration of dynein was sufficiently low for visualisation of individual binding events on microtubules.

We first asked whether, in conjunction with EB1, the native dynactin complex is sufficient to induce the association of dynein with the plus ends of growing microtubules (*Figure 2*). EB1 tracked the microtubule plus ends throughout growth phases in our assay conditions, as evidenced by its ability to induce this behaviour in a GFP-labelled binding partner, CLIP-170 (*Figure 2—figure supplement 1A*). The functionality of our EB1 preparation was also supported by its stimulation of microtubule catastrophe (*Figure 2—figure supplement 1B*), a property of EB1 reported in several previous studies (*Bieling et al., 2007*; *Komarova et al., 2009*; *Li et al., 2012*; *Vitre et al., 2008*). In the presence of both EB1 and the native dynactin complex, which was purified from pig brain, a subset of microtubule-associated Alexa647-dynein complexes exhibited binding events that followed the trajectory of the growing plus end for ≥1.8 s (*Figure 2A,B*). Such behaviour was operationally defined as plus-end tracking. The mean and maximal duration of dynein tracking events at the plus end in these conditions was ~4 and ~9 s, respectively (*Figure 2C,D*). Tracking behaviour was almost never observed when dynein was mixed with only EB1 or only dynactin (*Figure 2A,B*). Thus, EB1 and the full dynactin complex can act together to promote association of dynein with the plus ends of growing microtubules.

As described in the Introduction, LIS1 is required for association of dynein with plus ends of growing microtubules in mammalian cells (*Splinter et al., 2012*). We therefore investigated if LIS1 can enhance the association of mammalian dynein with growing microtubule plus ends in a minimal in vitro system. It was previously shown that LIS1 cannot autonomously bind microtubule plus ends (*Coquelle et al., 2002*; *Tai et al., 2002*). Consistent with this finding, hardly any dynein tracking events were observed when the motor complex was mixed with LIS1 alone or LIS1 and dynactin (*Figure 2A,B*). LIS1 did, however, strongly increase the number of plus-end tracking events induced by the combination of dynactin and EB1 (*Figure 2A,B*), without changing the duration of microtubule growth phases (*Figure 2—figure supplement 2A*). The duration of dynein tracking events on the growing plus end in the presence of EB1 and dynactin was also strongly increased by LIS1 (*Figure 2A,C,D*); the mean duration of tracking events increased by more than three-fold, with a subset of motor complexes remaining bound to the growing plus ends for longer than 30 s (*Figure 2C,D*). We conclude from these experiments that LIS1 strongly increases the frequency and duration of tracking events of dynein on the plus end of growing microtubules in conjunction with EB1 and dynactin.

We next investigated if LIS1 increases the frequency of plus-end tracking events by dynein at the growing plus end by promoting the initial landing of the motor complex at this site. In order to do this, we compared the incidence of dynein complexes landing on the growing plus end vs an equivalently-sized site on the GDP lattice. The presence of both EB1 and dynactin in the chamber made dynein approximately twice as likely to land at the plus-end site than at the lattice site (*Figure 2—figure supplement 2B*). The addition of LIS1 to these proteins did not further increase the frequency of dynein complexes landing on the plus end vs the lattice site (*Figure 2—figure supplement 2B*). Collectively, these data indicate that LIS1 increases the frequency of tracking events by prolonging the association of dynein with the growing plus end after it has been targeted to this site by EB1 and dynactin.

Interestingly, in the same assays in which LIS1 increased the duration of plus-end tracking events by dynein in the presence of EB1 and dynactin, there was no significant change in the duration of lattice binding events (*Figure 2E*). Thus, LIS1 specifically increased the persistence of EB1- and dynactin-dependent binding of dynein at the plus end of growing microtubules. To determine if this effect was due to preferential association of LIS1 with dynein at the growing plus end, co-localisation experiments were performed with Alexa647-dynein and Tetramethyl rhodamine (TMR)-labelled LIS1 in the presence of dynactin and EB1. TMR-LIS1 was detected with the vast majority of Alexa647-dynein complexes tracking the growing plus end (*Figure 2F*, *Figure 2—figure supplement 3A*), suggesting that LIS1 directly regulates dynein behaviour at this site. TMR-LIS1 was also detected on a large proportion of dynein complexes that were associated with the lattice (*Figure 2F*, *Figure 2—*

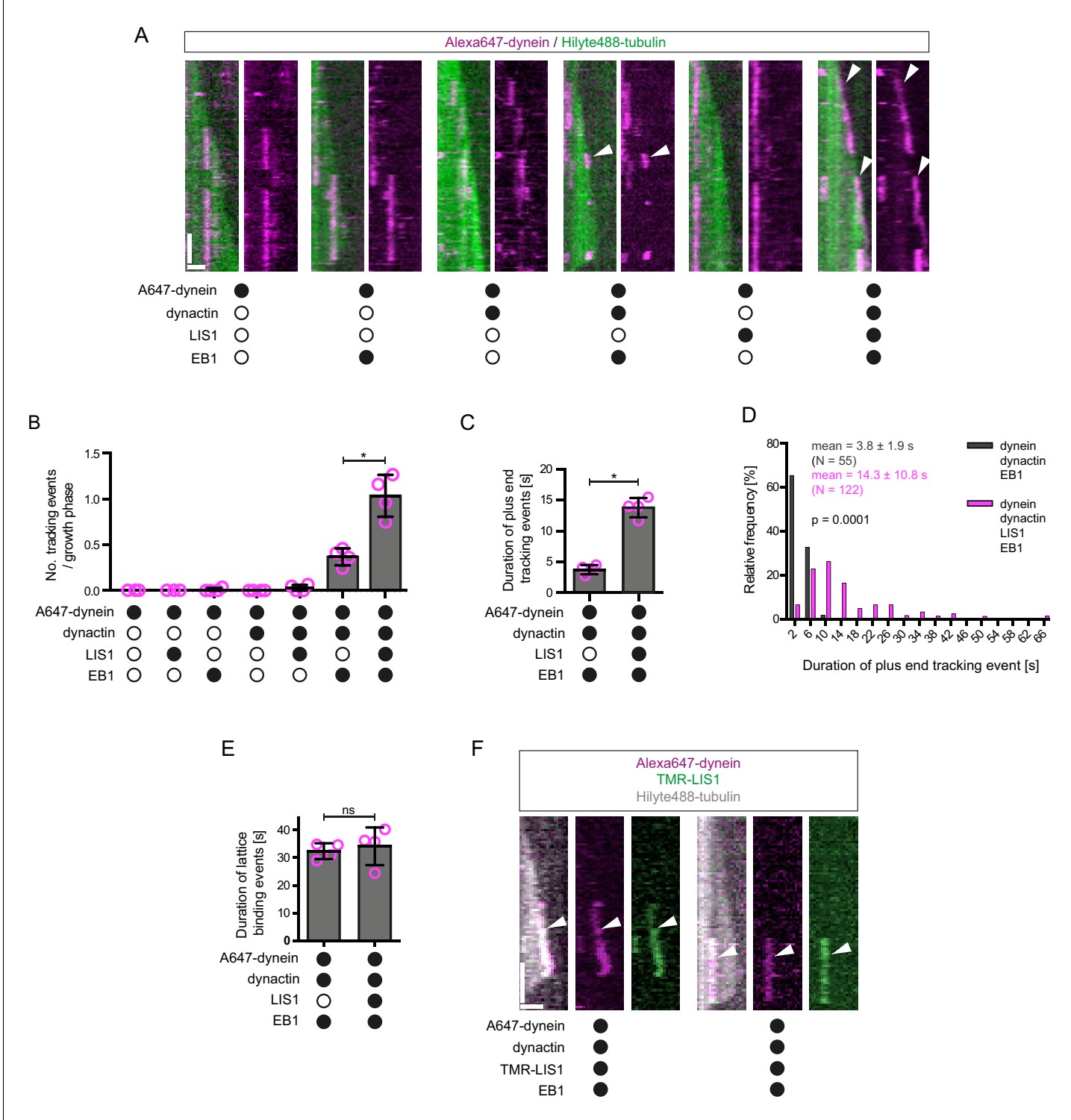

**Figure 2.** LIS1 induces persistent association of dynein complexes with the plus end of growing microtubules in the presence of EB1 and dynactin. (A) Kymographs showing examples of dynein behaviour on growing microtubules in the presence of the indicated proteins (filled circles). In **A** and **F**: Y-axis, time; x-axis, distance; scale bars, 10 s and 1 μm. In these and other kymographs, the microtubule plus end is to the right. Arrowheads, plus-end tracking events. As expected, minus end-directed dynein transport was hardly ever observed in these conditions or others without either dynactin or BICD2N. (B–D) Quantification of LIS1's effects on the number (B) and duration (C–D) of dynein tracking events at the plus end during growth phases and (E) the duration of dynein binding events on lattice sites (including the GMPCPP seed). (F) Kymographs showing examples (arrowheads) of TMR-LIS1 localisation with Alexa647-dynein that undergoes tracking on growing microtubule plus ends (left kymograph) or lattice binding (right kymograph).

*Figure 2 continued on next page*

*Figure 2 continued*

LIS1 was labelled with a C-terminal SNAP$_f$ tag. See *Figure 2—figure supplement 3* for quantification of co-localisation in multiple chambers. In B, C and E, means ± S.D. are shown with values for each chamber represented as magenta circles (four chambers per condition, except for dynein alone and dynein plus LIS1 in B) (three chambers); mean of 35 growth phases (B), 22 events (C–D) or 87 events (E) analysed per chamber. (D) shows overall distribution of the duration of dynein's plus-end tracking events during growth phases (errors are S.D.; N is total number of tracking events summed from four chambers per condition). Statistical significance in B–E was evaluated with a Mann-Whitney test (*p<0.05; ns, not significant). Dynactin complexes and LIS1 dimers were used, respectively, at a molar excess of 2x and 20x that of dynein. The concentration of the dynein complex in the assembly mixes was 20 nM, with a 1 in 10 dilution added to the imaging chamber. In these and other experiments EB1 dimers were present in the imaging chamber at a concentration of 100 nM.

The following figure supplements are available for figure 2:

**Figure supplement 1.** Further evidence of EB1 functionality.

**Figure supplement 2.** Additional data on microtubule dynamics and the landing site of dynein on growing microtubules.

**Figure supplement 3.** Additional data on dynein and LIS1 co-localisation on dynamic microtubules.

*figure supplement 3B*). We conclude that LIS1 can associate with dynein at the growing plus end and on the lattice, but increases the duration of microtubule binding events only at the former site.

The observation that LIS1 does not influence the dwell time of human dynein on the microtubule lattice was unexpected as yeast LIS1 strongly increases the dwell time of monomeric yeast dynein on stabilised microtubules in the presence of ATP (*Huang et al., 2012*). Moreover, it has been reported using a bulk co-sedimentation approach that LIS1 can increase the association of porcine or bovine dynein with stabilised microtubules, which lack a dynamic plus end, in the absence of EB1 and dynactin (*McKenney et al., 2010*; *Yamada et al., 2013*). To attempt to reconcile our findings with those of the previous studies, we investigated in more detail the influence of LIS1 on the interaction of single human dynein complexes with the microtubule lattice in the absence of EB1 and dynactin. These experiments used microtubules that were stabilised with taxol and GMPCPP, which facilitated the analysis of binding events by dynein. Using the same relative concentration of LIS1 to dynein used in our dynamic microtubule assays, we again detected no influence of LIS1 on the dwell time of the motor complex on the lattice (*Figure 3A,B*). The rate at which dynein complexes landed on microtubules was also not significantly influenced by LIS1 in these experiments (*Figure 3A,C*). Even a 300-fold molar excess of LIS1 to dynein did not increase the dwell time of the motor complex on microtubules (*Figure 3A,B*, *Figure 3—figure supplement 1*). However, in this condition, there was an approximately three-fold increase in dynein's landing rate on the lattice (*Figure 3A,C*). Thus, a high relative concentration of LIS1 to dynein can increase the association of single motor complexes with microtubules by increasing the on-rate. This finding is compatible with the previous observations that LIS1 promotes the co-sedimentation of other mammalian dyneins with microtubules. The basis of the concentration-dependent effects of LIS1 on the association rate of human dynein with microtubules is unclear but could conceivably be related to more than one binding site for LIS1 on the dynein complex (see Discussion). Collectively, our data suggest that, unlike the situation with yeast proteins, human LIS1 is not sufficient to increase the dwell time of individual human dynein complexes on microtubules. LIS1 can, however, increase the dwell time of human dynein complexes at the plus ends of growing microtubules, which are recruited through EB1 and dynactin.

## EB1 can direct dynein-dynactin-BICD2N transport initiation to growing plus ends, but this is not influenced by LIS1 or CLIP-170

The experiments described above shed light on how dynein interacts with the plus ends of growing microtubules in the absence of activators of processive minus end-directed movement. In order to investigate how mammalian dynein complexes that undergo minus end-directed motion are targeted to dynamic microtubules, both dynactin and the N-terminal region of BICD2 (BICD2N) were included in the in vitro assay. This region of BICD2 is sufficient to associate with dynein and dynactin (*Splinter et al., 2012*) and stimulate processive movement (*McKenney et al., 2014*; *Schlager et al., 2014*). In the absence of EB1, minus end-directed movements of dynein-dynactin-BICD2N

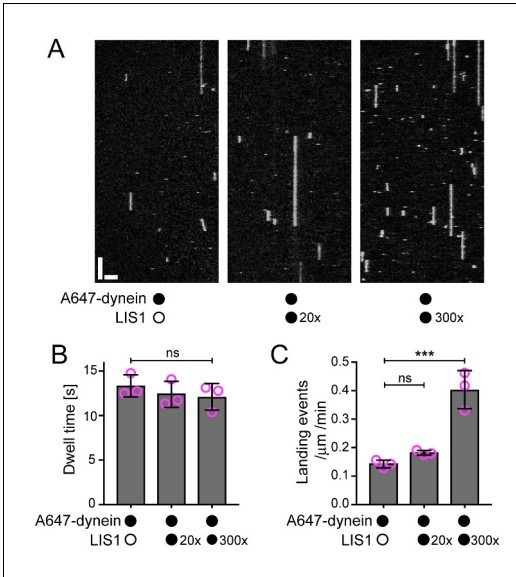

**Figure 3.** Evaluation of the influence of LIS1 on the interaction of individual human dynein complexes with microtubules in the absence of other factors. (**A**) Kymographs showing examples of dynein behaviour on taxol/GMPCPP-stabilised microtubules in the presence of the stated molar excess of LIS1 to dynein. Y-axis, time; x-axis, distance; scale bars, 10 s and 2 μm (**B**) Quantification of the effect of LIS1 on dynein dwell time (**B**) and landing rate (events per μm of microtubule per minute) (**C**). Means ± S.D. are shown with values for each chamber represented as magenta circles (three chambers per condition; mean of 28 microtubules and 210 complexes (no LIS1 and 20x LIS1) or 545 complexes (300x LIS1) analysed per chamber). Statistical significance was evaluated with a one-way ANOVA with Sidak's multiple comparisons test (***p<0.001; ns, not significant). Dynein concentration in the assembly mixes was 100 nM, with a 1 in 20 dilution added to the imaging chambers.

The following figure supplement is available for figure 3:

**Figure supplement 1.** Distribution of dwell times of individual dynein complexes on stabilised microtubules in the presence and absence of LIS1.

complexes were just as likely to initiate from the plus end as they were from a randomly selected segment of the GDP lattice of the same length (*Figure 4A,B*). Strikingly, addition of EB1 resulted in minus end-directed transport events being six times more likely to initiate at the plus-end location (*Figure 4A,B*). The velocity and run length of processive dynein-dynactin-BICD2N movements in the presence of EB1 did not differ significantly depending on an initiation site on the lattice vs at the plus end (*Figure 4—figure supplement 1A, B*). Thus, EB1 can direct the site of dynein-dynactin-BICD2N transport initiation to the plus end of growing microtubules but this does not influence the motile properties of the complex.

It has been proposed that LIS1 regulates initiation of dynein-based cargo transport from microtubule plus ends in at least some contexts (*Egan et al., 2012*; *Lammers and Markus, 2015*; *Lenz et al., 2006*). We therefore investigated whether LIS1 can modulate EB1's ability to target initiation of dynein-dynactin-BICD2N transport events to the plus end of growing microtubules. In the presence of EB1, dynactin, BICD2N and LIS1, we again observed a six-fold higher frequency in the initiation of processive dynein movements from the plus end vs an equivalently-sized GDP lattice site (*Figure 4A,B*). Thus, LIS1 did not alter EB1's ability to bias transport initiation to the plus end. LIS1 did, however, stimulate both the frequency and duration of plus-end tracking events on growing microtubules in the presence of EB1, dynactin and BICD2N, without affecting the duration of static binding events by the motor on the lattice (*Figure 4—figure supplement 1C–G*). No end tracking of dynein was observed when the motor complex was mixed with only EB1, BICD2N and LIS1 (32 growth phases analysed), confirming the essential role of dynactin in inducing this behaviour. Thus, LIS1 can promote EB1- and dynactin-dependent tracking of dynein on growing plus ends in the presence and absence of BICD2N in the assay chamber.

CLIP-170 can contact EB1, LIS1 and p150 and plays an important role in enriching dynein at microtubule plus ends in vivo (*Lansbergen et al., 2004*; *Watson and Stephens, 2006*). We therefore asked if, in vitro, CLIP-170 can stimulate the EB1-mediated targeting of processive dynein-dynactin-BICD2N complexes to growing plus ends in the presence of LIS1. The inclusion of both CLIP-170 and LIS1 did not alter the site of transport initiation of dynein-dynactin-BICD2N complexes in the presence of EB1 (*Figure 4—figure supplement 2A*). Furthermore, CLIP-170 did not affect the frequency or duration of plus-end tracking events of dynein observed in the presence of EB1, dynactin, BICD2N and LIS1 (*Figure 4—figure supplement 2B–E*). This latter finding is consistent with the observation that CLIP-170 does not influence the EB1-mediated plus-end tracking of a truncated p150 isoform in vitro unless competing EB1-binding peptides are present (*Duellberg et al., 2014*).

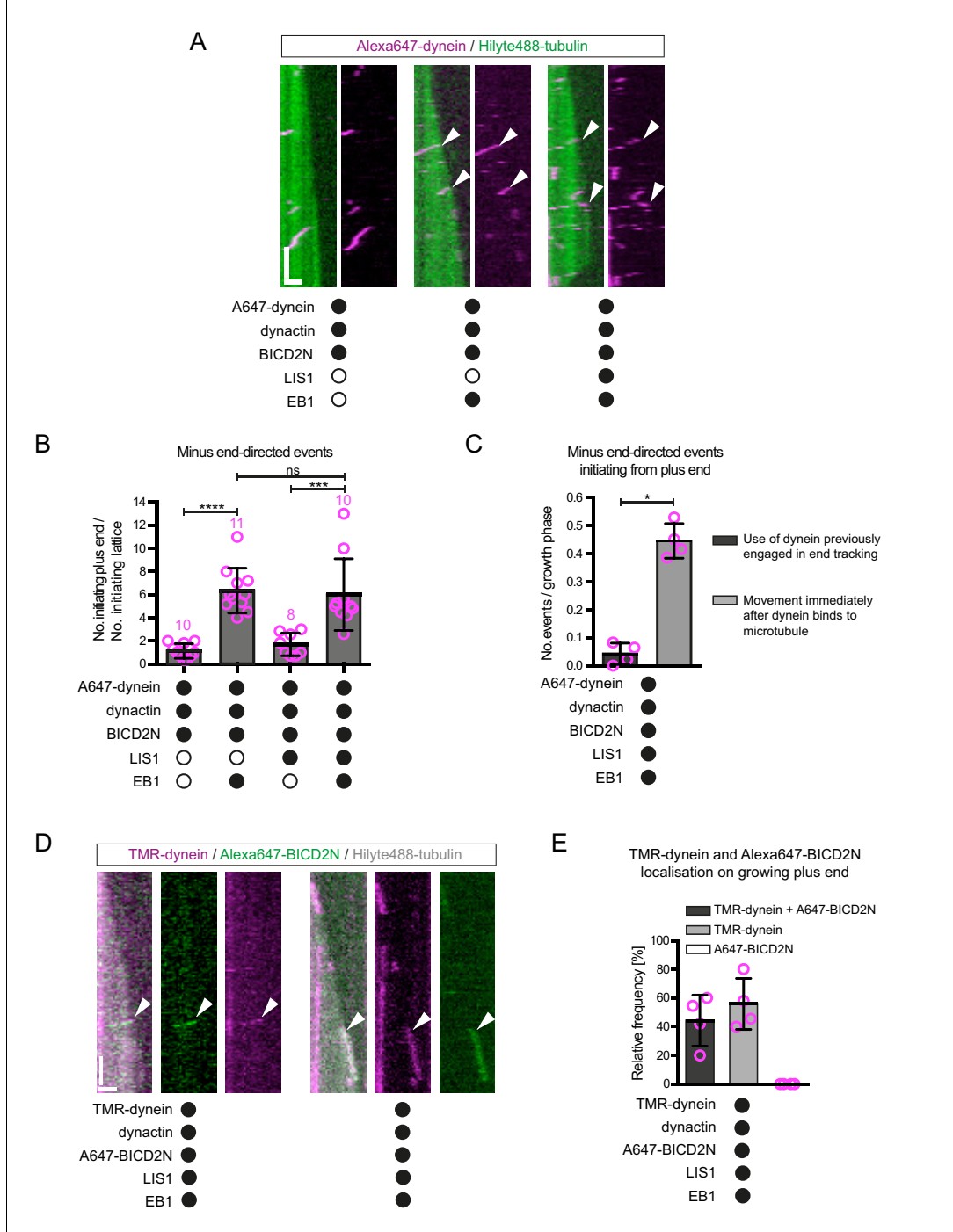

**Figure 4.** The effect of EB1 and LIS1 on the initiation site of minus end-directed transport of dynein in the presence of dynactin and BICD2N. (A) Kymographs showing examples of dynein behaviour on polymerising microtubules. Arrowheads, examples of initiation of minus end-directed motion of dynein from the growing plus end in the presence of dynactin, BICD2N and EB1. In these and other experiments, BICD2N has an N-terminal SNAP$_f$ tag. In A and D: y-axis, time; x-axis, distance; scale bars, 10 s and 1 μm. (B) Quantification of initiation of minus end-directed motility of dynein on the growing plus end vs a site on the GDP lattice of the same length. (C) Quantification of the source of dynein used for minus end-directed motion of dynein-dynactin-BICD2N complexes on dynamic microtubules. (D) Kymographs of growing microtubules showing that Alexa647-BICD2N can be detected with TMR-dynein engaged in minus end-directed transport (left kymograph, arrowhead) or end tracking (right kymograph, arrowhead). See *Figure 4—figure supplement 3B* for another example of BICD2N localisation on minus end-directed dynein. BICD2N was labelled using the SNAP$_f$ tag. (E) Quantification of the proportion of plus-end tracking events of TMR-dynein or Alexa647-BICD2N on growing microtubules with detectable signals from both proteins. In B, C and E, means ± S.D. are shown with values for each chamber represented as magenta circles (four chambers per condition, except in B) where the number of chambers is shown in magenta above the bars; mean of 22 complexes (B), 33 growth phases (C) or 17

*Figure 4 continued on next page*

*Figure 4 continued*

complexes (**E**) analysed per chamber). Statistical significance was evaluated with a one-way ANOVA with Sidak's multiple comparisons test (**B**) or a Mann-Whitney test (**C**) (****$p<0.0001$; ***$p<0.001$; *$p<0.05$; ns, not significant). Dynactin complexes, BICD2N dimers and LIS1 dimers were used, respectively, at a molar excess of 2x, 10x and 20x that of dynein. Dynein concentration in the assembly mixes was 20 nM, with a 1 in 10 (**A**–**C**) or 1 in 2 (**D**,**E**) dilution added to the imaging chambers.

The following figure supplements are available for figure 4:

**Figure supplement 1.** Additional data on the behaviour of dynein on dynamic microtubules in the presence of BICD2N and other regulators.

**Figure supplement 2.** Evaluation of the effects of CLIP-170 on dynein behaviour in the presence of dynactin, BICD2N, LIS1 and EB1.

**Figure supplement 3.** Further analysis of the co-localisation of BICD2N and dynein signals on dynamic microtubules.

We conclude that EB1, but not LIS1 or CLIP-170, is a key factor in targeting processive dynein-dynactin-BICD2N complexes to the plus ends of growing microtubules in our assay system.

## End tracking dynein complexes are rarely a source of motor for minus end-directed movements

It has been hypothesised that tracking of dynein on the plus ends of growing microtubules in vivo produces a loading zone for cargo that is destined for minus end-directed transport (*Lenz et al., 2006*; *Lloyd et al., 2012*; *Vaughan, 2004*; *Vaughan et al., 1999*, *2002*). The finding that LIS1 can strongly increase the frequency and duration of dynein tracking events at the plus end without increasing the initiation of minus end-directed movements from this site suggests that such a mechanism is not at work in our experimental system. Indeed, further analysis revealed that in the presence of EB1, dynactin, BICD2N and LIS1, only 10% of minus end-directed movements from the plus end initiated from a plus-end tracking event of dynein (*Figure 4C*). Instead, minus end-directed movements from the plus end almost always occurred immediately after engagement of the motor complex with this site (*Figure 4C*). This was also the case when CLIP-170 was included in the assay (*Figure 4—figure supplement 2B*). Thus, dynein complexes that track the plus ends of microtubules rarely act directly as a source of motors for minus end-directed transport in our assay conditions.

Plus-end tracking events may not be converted into minus end-directed transport events because dynein complexes are in a state that cannot accommodate BICD2N. To test this possibility, we produced Alexa647-labelled BICD2N and analysed its localisation on microtubules in the presence of EB1, dynactin, LIS1 and TMR-labelled dynein. As expected, Alexa647-BICD2N was detected on almost all minus end-directed TMR-dynein complexes (*Figure 4—figure supplement 3A*), including those that initiated movement immediately after engaging with the growing plus end (*Figure 4D*, *Figure 4—figure supplement 3B*). Alexa647-BICD2N was also detected on ~40% of events in which TMR-dynein tracked the growing plus ends of microtubules (*Figure 4D,E*). The dual-labelled complexes that tracked the plus end never gave rise to minus end-directed transport events; all 38 dual-labelled complexes that initiated minus end-directed transport from the plus end did so immediately after docking at this site. Because dynein cannot track growing plus ends without dynactin, we conclude that the end tracking complexes containing Alexa647-BICD2N and TMR-dynein are dynein-dynactin-BICD2N complexes that are not capable of minus end-directed movement. The frequency of co-localisation of BICD2N with dynein complexes engaged in plus-end tracking (*Figure 4E*) was similar to that observed for static dynein complexes on the lattice (*Figure 4—figure supplement 3C*). This finding raises the possibility that end tracking behaviour of dynein-dynactin-BICD2N complexes results from the recruitment of the non-processive subset of these complexes to the plus end, and that additional factors are required for transport of end tracking dynein-dynactin-cargo adaptor complexes from this site in vivo (see Discussion).

## LIS1 increases the frequency and velocity of minus end-directed dynein movements

Our earlier results showed that LIS1 does not regulate the site of initiation of minus end-directed dynein-dynactin-BICD2N movements on dynamic microtubules. We next asked if LIS1 has any

influence on the motility of dynein along microtubules in the presence of dynactin and BICD2N. Interestingly, LIS1 increased the proportion of microtubule-associated dynein complexes that underwent minus end-directed motion from 45% to 70% (*Figure 5A*). When the relative concentration of BICD2N was reduced 10-fold, we observed hardly any dynein complexes exhibiting minus end-directed transport in the absence of LIS1 (*Figure 5B,C*). In the presence of LIS1, however, a sizeable fraction of dynein complexes were transported towards the microtubule minus end (*Figure 5B,C*). Thus, LIS1 can induce dynein transport with dynactin and BICD2N concentrations that are otherwise insufficient to elicit this behaviour. We conclude that although LIS1 is not essential for minus end-directed movements of dynein when incubated with dynactin and BICD2N, it can strongly stimulate the frequency of these events.

We and others have demonstrated that LIS1 can promote the association of dynein with dynactin in cells or cell extracts (*Dix et al., 2013*; *Wang et al., 2013*), although it was not known if other factors are involved in this process. We hypothesised that LIS1 promotes minus end-directed motion of dynein in the presence of dynactin and BICD2N in vitro by stimulating formation of the dynein-dynactin-BICD2N complex. Quantification of fluorescent signals on microtubules revealed that LIS1 strongly increased the association of Alexa647-BICD2N with TMR-dynein in the presence of dynactin (*Figure 5B,D*). Coupled to previous reports that dynactin is required for association of BICD2N with dynein (*Hoang et al., 2017*; *McKenney et al., 2014*; *Schlager et al., 2014*; *Splinter et al., 2012*), these data indicate that LIS1 promotes the assembly of the dynein-dynactin-BICD2N complex. This ability of LIS1 offers an explanation for how it stimulates processive motion of dynein in the presence of BICD2N and dynactin.

In the presence of LIS1, the mean and maximum velocity of minus end-directed dynein-dynactin-BICD2N complexes on dynamic microtubules also increased markedly, regardless of whether EB1 was included in the assay (*Figure 6A,B*, *Figure 6—figure supplement 1*). LIS1 did not, however, strongly affect the run length of motile dynein-dynactin-BICD2N complexes (*Figure 6C*). The ability of LIS1 to increase the velocity of dynein-dynactin-BICD2N complexes was not restricted to microtubules with a dynamic plus end, as a strong effect was also observed in experiments with stabilised microtubules (*Figure 6D*). Increased velocity of dynein-dynactin-BICD2N movement was observed over a large range of LIS1 protein concentrations (*Figure 6D*).

Some studies have proposed that the copy number of dyneins bound to a physiological cargo can regulate velocity in vivo, with higher motor numbers associated with higher speeds (*Kural et al., 2005*; *Levi et al., 2006*). We used the fluorescent intensities of Alexa647-dynein to investigate if the increase in dynein-dynactin-BICD2N velocity induced by LIS1 correlates with an increase in the copy number of the motor in minus end-directed complexes. There was no significant difference in the mean fluorescent intensity of dynein within minus end-directed transport complexes in the presence and absence of LIS1 (*Figure 6E*). Moreover, there was no correlation between the fluorescence intensity of transported dynein complexes and their velocity (*Figure 6—figure supplement 2*). These experiments suggest that multimerisation of the motor does not account for LIS1's ability to stimulate dynein velocity.

The strong stimulation of dynein-dynactin-BICD2N velocity by LIS1 was surprising as previous in vitro studies have shown that LIS1 reduces the activity of isolated dynein complexes (*Huang et al., 2012*; *McKenney et al., 2010*; *Torisawa et al., 2011*; *Toropova et al., 2014*; *Wang et al., 2013*; *Yamada et al., 2008*). For instance, LIS1 strongly inhibits microtubule gliding by purified yeast (*Huang et al., 2012*), porcine (*Torisawa et al., 2011*; *Yamada et al., 2008*) or bovine (*Wang et al., 2013*) dynein complexes that are immobilised on a glass surface. We therefore tested whether our purified LIS1 inhibited microtubule gliding by immobilised human recombinant dynein. As documented previously (*Hoang et al., 2017*; *Schlager et al., 2014*; *Trokter et al., 2012*), human recombinant dyneins adsorbed on a glass surface induced robust gliding of microtubules in the presence of ATP (*Figure 6F*, *Figure 6—figure supplement 3A*, *Video 1*). Addition of LIS1 to dynein during the surface adsorption step did not inhibit the subsequent association of microtubules with the glass surface but did result in a dose-dependent inhibition of microtubule translocation (*Figure 6F*, *Figure 6—figure supplement 3A*, *Video 1*). As microtubules do not adhere to the surface in the absence of dynein in these assay conditions (*Hoang et al., 2017*), we reasoned that LIS1 does not inhibit microtubule gliding by displacing the motor complex from the glass. Confirming this notion, LIS1 could strongly inhibited microtubule gliding without reducing the amount of fluorescently labelled dynein on the glass surface (*Figure 6—figure supplement 3B*). We also confirmed that

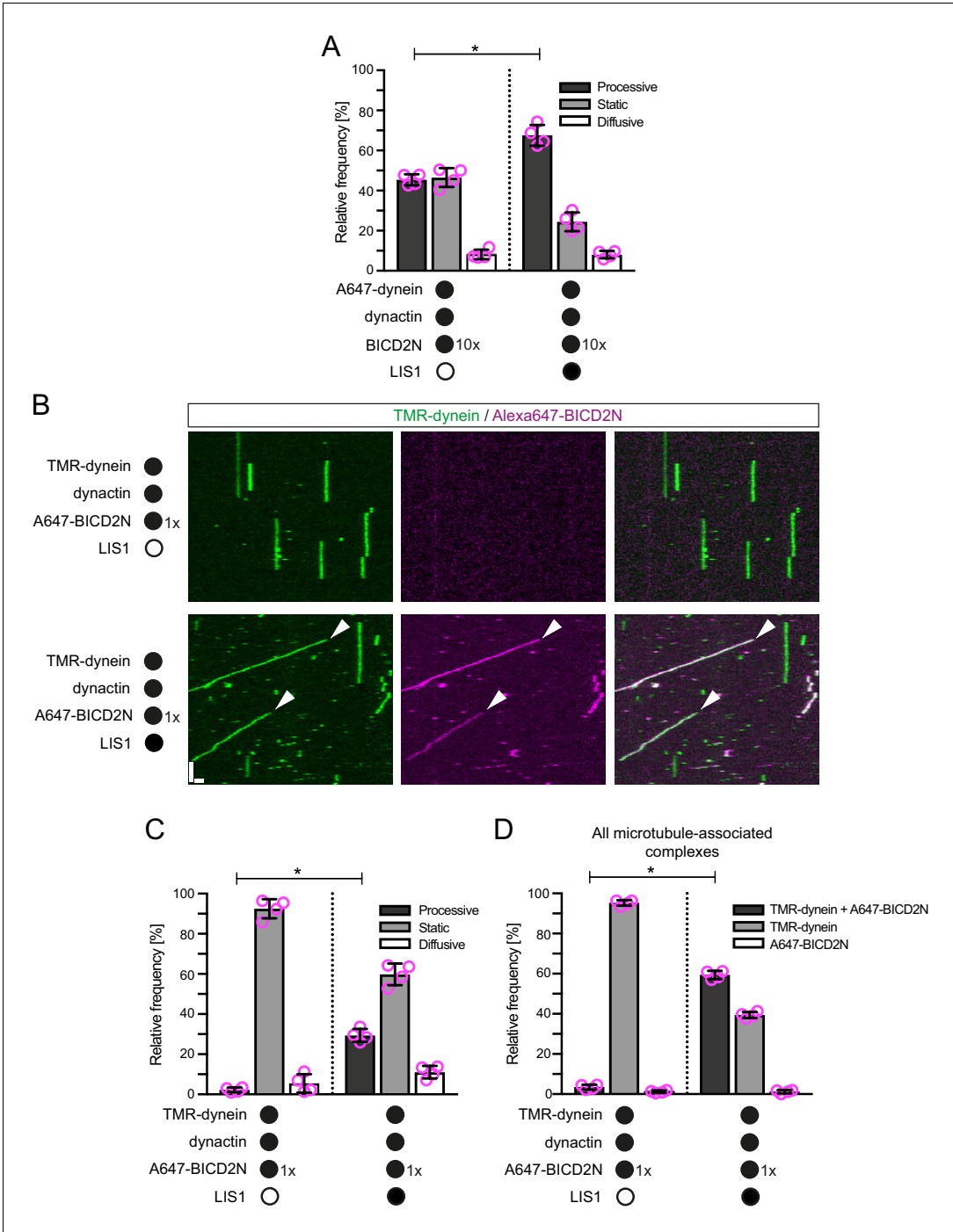

**Figure 5.** LIS1 promotes the frequency of processive minus end-directed movement of dynein in the presence of dynactin and BICD2N. (**A**) Quantification of the effect of LIS1 on the motile behaviour of microtubule-associated dynein complexes in the presence of BICD2N and dynactin. Dynactin complexes, BICD2N dimers and LIS1 dimers were used, respectively, at a molar excess of 2x, 10x and 20x compared to dynein. (**B**) Kymographs from experiments in which the relative concentration of BICD2N is lowered 10-fold. Arrowheads show examples of processive minus end-directed dynein-dynactin-BICD2N complexes in the presence of LIS1. Y-axis, time; x-axis, distance; scale bars, 10 s and 1 μm. (**C–D**) Quantification of the effect of LIS1 on the motile behaviour (**C**) or association with BICD2N (**D**) of microtubule-associated dynein complexes in the presence of dynactin and the lower relative concentration of BICD2N. **A** and **B–D** used dynamic and stabilised microtubules, respectively. In **A**, **C** and **D**, means ± S.D. are shown with values for each chamber represented as magenta circles (four chambers per condition; mean of 177 (**A**) or 249 (**C–D**) complexes analysed per chamber). Statistical significance was

*Figure 5 continued on next page*

*Figure 5 continued*
evaluated with a Mann-Whitney test (\*p<0.05). Dynein concentration in the assembly mixes was 20 nM (with a 1 in 10 dilution added to the imaging chamber) (**A**) or 100 nM (with a 1 in 40 dilution added to the imaging chambers (**B–D**)).

microtubules did not stably associate with glass surfaces incubated with only LIS1 (*Video 2*). This finding corroborates the results of previous studies showing that LIS1 does not directly interact with microtubules (*McKenney et al., 2010*; *Yamada et al., 2008*). Thus, LIS1 does not inhibit microtubule gliding by human dynein indirectly by providing a strong independent attachment between microtubules and the glass surface. We conclude that LIS1 can directly inhibit the ability of human recombinant dynein to glide microtubules. Thus, LIS1 exerts different functional effects on ensembles of isolated human dynein complexes in a gliding assay and on individual dynein-dynactin-BICD2N complexes running along immobilised microtubules.

## LIS1 associates with minus end-directed dynein-dynactin-BICD2N complexes

In vivo work in fungal systems has provided compelling evidence that LIS1 dissociates from dynein cargoes before transport begins (*Egan et al., 2012*; *Lammers and Markus, 2015*; *Lenz et al., 2006*). We therefore investigated if LIS1 is present on motile dynein-dynactin-BICD2N complexes in our assay (*Figure 7*). Motility assays on dynamic microtubules were performed in the presence of TMR-LIS1, Alexa647-dynein, dynactin and BICD2N. In addition to frequently associating with Alexa647-dynein complexes that tracked the growing plus end or exhibited non-processive behaviour on the lattice (*Figure 7—figure supplement 1*), TMR-LIS1 was co-transported with the vast majority of processive Alexa647-labelled dynein-dynactin-BICD2N complexes (*Figure 7A,B*). The mean velocity of minus end-directed dynein complexes with a TMR-LIS1 signal (*Figure 7C*) was similar to that observed in the previous experiments using unlabelled LIS1 (*Figure 6B*). Within the same imaging chambers, the subset of minus end-directed Alexa647-dynein complexes that did not have a TMR-LIS1 signal moved with a significantly reduced speed compared to those that did (*Figure 7C*). These data indicate that LIS1 regulates the velocity of dynein-dynactin-BICD2N transport complexes by associating with them.

## Dynein can remain bound to the plus end of microtubules during shrinkage and LIS1 can promote this behaviour

The experiments described above were designed to shed light on how dynein complexes interact with microtubules during bouts of growth. However, we were also able to observe the behaviour of dynein on the plus end of microtubules when growth phases switched to shrinkage phases. Many dyneins that tracked the plus end during a growth phase did not dissociate from this site when a shrinkage event began and instead retreated with the depolymerising end (*Figure 8A,B*). Dynein bound to a lattice site could also retreat with the shrinking end when the depolymerisation process swept through its binding site, including in the absence of EB1 (*Figure 8A*). The association of dynein with shrinking microtubule ends could last for several seconds, during which time hundreds of tubulin subunits dissociate from the plus end (*Karr et al., 1980*). We subsequently realised that dynein appeared to track the shrinking ends of microtubules in an assay of *Duellberg et al. (2014)* that contained EB1 and the truncated p150 isoform (left panel in their Video 3). However, the underlying mechanism was not investigated.

Our finding that EB1 is not needed for association of dynein with the plus end of shrinking microtubules is consistent with the observation that EB family members only track plus ends during growth phases (*Bieling et al., 2008*, *2007*; *Maurer et al., 2012*). We confirmed that EB proteins do not track shrinking ends in our assay conditions, including those bound to dynein (*Figure 8C*). This was achieved using an mCherry-tagged EB3 protein (*Duellberg et al., 2014*; *Montenegro Gouveia et al., 2010*), which induced tracking of dynein on growing plus ends in the presence of dynactin and LIS1 (*Figure 8—figure supplement 1A*). These data indicate that although an EB protein is required for dynein to track the plus end during growth, this is not the case during shrinkage. Interestingly, we never observed dynein tracking on a shrinking microtubule plus end in the absence of

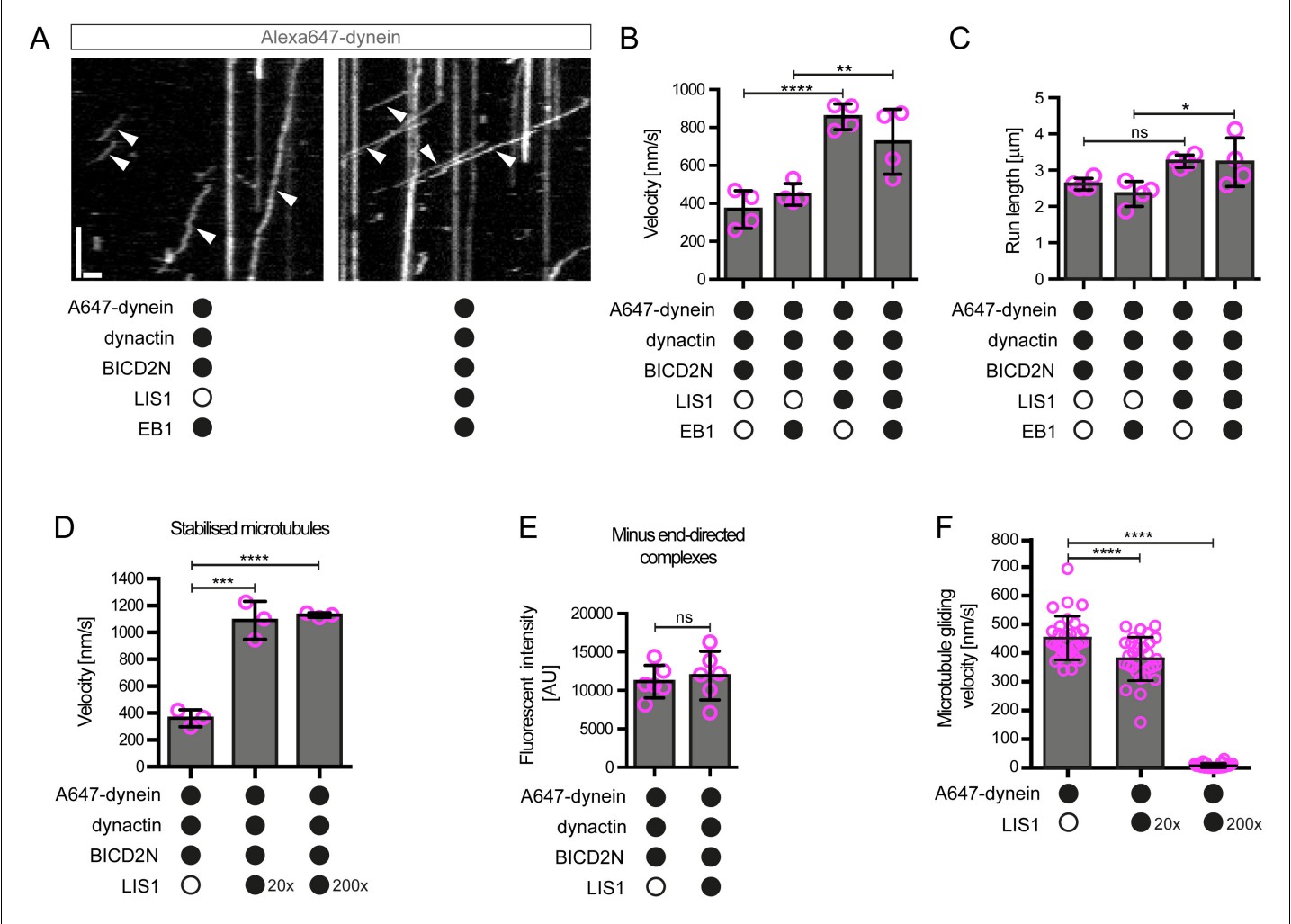

**Figure 6.** LIS1 increases the velocity of minus end-directed movements of dynein in the presence of dynactin and BICD2N. (**A**) Kymograph from experiments with dynamic microtubules illustrating increased velocity of minus end-directed dynein-dynactin-BICD2N complexes in the presence of LIS1. Arrowheads: examples of minus end-directed complexes. Y-axis, time; x-axis, distance. Scale bars, 3 s and 1 µm. (**B–C**) Quantification of the velocity (**B**) and run length (**C**) of minus end-directed dynein-dynactin-BICD2N complexes on dynamic microtubules in the presence and absence of LIS1. See *Figure 6—figure supplement 1* for examples of velocity distributions. (**D**) Quantification of minus end-directed velocity of dynein-dynactin-BICD2N complexes on taxol/GMPCPP-stabilised microtubules in the presence and absence of LIS1. (**E**) Quantification of fluorescence intensity of Alexa647-dynein in minus end-directed complexes in the presence and absence of LIS1. The experiments were performed with stabilised microtubules. (**F**) Quantification of LIS1's inhibitory effect on microtubule gliding by surface immobilised human dynein. Dynein alone or with a 20x or 200x molar excess of LIS1 dimers was mixed and incubated with glass surfaces, followed by washing and addition of fluorescent microtubules. In **B–D**, means ± S. D. are shown with values for each chamber represented as magenta circles (four chambers per condition in **B** and **C**, three chambers per condition in **D**); mean number of complexes analysed per chamber: 60 (**B, C**) and 106 (**D**). In **E**, means ± S.D. are shown with values for each movie represented as magenta circles (six movies from two chambers per condition, with a mean of 102 complexes analysed per chamber). In **F**, means ± S.D. are shown with values for each microtubule represented as magenta circles (30 microtubules from two chambers per condition). Statistical significance was evaluated with a one-way ANOVA with Sidak's multiple comparisons test (**B–D** and **F**) or a Mann-Whitney test (**E**). (****$p<0.0001$; *** $p<0.001$; **$p<0.01$; *$p<0.05$; ns, not significant). Dynactin complexes, BICD2N dimers and LIS1 dimers were used, respectively, at a molar excess of 2x, 10x and 20x compared to dynein, except in some experiments in **D** and **F**, when LIS1 dimers were included at a molar excess of 200x relative to dynein. Dynein concentration in the assembly mixes was 20 nM (with a 1 in 10 dilution added to the imaging chambers) (**A–E**) or 30 nM (**F**). The acquisition rate was 7.3 frames/s in **A** and **D**, one frame/s in **F**, and 1.7 frame/s in other panels.

The following figure supplements are available for figure 6:

**Figure supplement 1.** Examples of velocity distributions from the data in *Figure 6B*.

*Figure 6 continued on next page*

*Figure 6 continued*

**Figure supplement 2.** Plot of fluorescent intensity vs velocity for minus end-directed Alexa647-dynein complexes on stabilised microtubules in the presence of dynactin, BICD2N and LIS1.

**Figure supplement 3.** Additional data on the effect of LIS1 on microtubule gliding by surface immobilised dynein.

dynactin (0/103 retention events following encounters of dynein with a shrinking end in the absence of dynactin (*Figure 8—figure supplement 1B*) vs 123/228 events in the presence of dynactin (*Figure 8B*)). Thus, dynactin appears to be required for the association of dynein with dynamic plus ends during bouts of shrinkage as well as growth.

We next asked whether LIS1 regulates association of dynein with the plus end of the microtubule during shrinkage. We first used fluorescently labelled proteins to determine if LIS1 is present on dynein that tracks shrinking ends. TMR-LIS1 was detected with Alexa647-dynein in 20 of 22 events analysed in which the motor remained associated with the plus end during shrinkage (*Figure 8—figure supplement 1C*). Further analysis revealed that, although LIS1 was not essential for maintenance of dynein on the plus end after encountering a shrinking end, it did significantly increase the frequency of this behaviour in combination with (i) dynactin, (ii) EB1 and dynactin and (iii) EB1, dynactin and BICD2N (*Figure 8B*). We did not, however, observe an altered frequency of shrinking end tracking events of dynein when LIS1 was added to a combination of dynactin and BICD2N (*Figure 8B*). This result raises the possibility that LIS1 has context-dependent effects on dynein's ability to track shrinking ends. Overall, however, our data provide evidence for a function of LIS1 in stabilising dynein complexes on the plus end of microtubules during shrinkage.

We next considered the possibility that dynein transitions to an active, minus end-directed stepping mechanism in order to remain attached to the plus end during shrinkage. To test this notion, we performed assays in the presence of ATP and vanadate. Vanadate leads to the formation of a stable complex (dynein-ADP·Vi), which mimics the ADP·Pi transition state and hence the pre-power-stroke conformation of the motor (*Burgess et al., 2003*; *Johnson, 1985*; *Shimizu and Johnson, 1983*). The experiment was performed in the presence of EB1, dynactin, BICD2N and LIS1, a condition in which tracking events on the shrinking end are frequently observed in the presence of ATP alone (*Figure 8B*). Vanadate strongly inhibited the minus end-directed transport of dynein-dynactin-BICD2N complexes (*Figure 8—figure supplement 1D*), providing an internal control for its inhibitory effect on ATP hydrolysis. In contrast, association of dynein with the plus ends of shrinking microtubules was frequently observed in the presence of vanadate (*Figure 8D*), with no statistically significant difference in the occurrence of these events compared to in the presence of only ATP (*Figure 8B*). We also noted that vanadate did not disrupt end tracking of the full dynein complex during growth phases (*Figure 8D*). Thus, the association of dynein with microtubule plus ends during both growth and shrinkage is not dependent on the ATP hydrolysis cycle.

## Discussion

We have used an in vitro reconstitution approach to shed light on how mammalian dynein complexes interact with microtubules undergoing growth and shrinkage. These experiments include an assessment of how purified dynein complexes behave on dynamic microtubules in the presence of activators of minus end-directed transport. Our results reveal a minimal set of interactions that govern association of end tracking and minus end-directed motor complexes with dynamic microtubules in the presence of the full dynactin complex, and reveal several

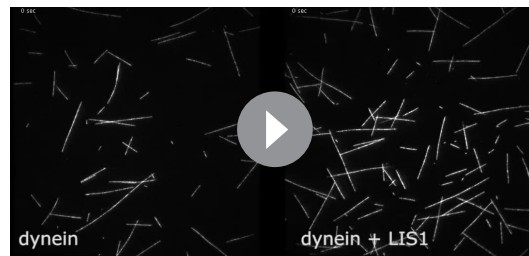

**Video 1.** LIS1 inhibits microtubule gliding by surface-immobilised human dynein complexes. Microtubules are labelled with HiLyte488-tubulin. Before addition of microtubules, glass surfaces were incubated with a solution of 30 nM dynein alone (left) or 30 nM dynein plus 6 μM LIS1 dimers and washed. Videos represents 5 min of real time; width of each frame is 54 μm.

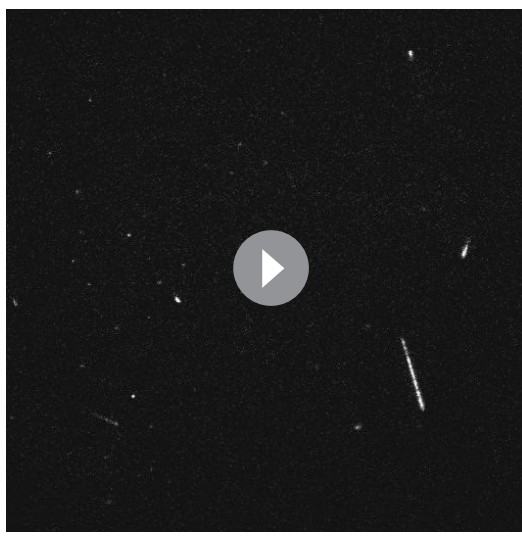

**Video 2.** Microtubules do not stably associate with glass surfaces incubated with LIS1 alone. Microtubules are labelled with HiLyte488-tubulin. Glass surfaces were pre-incubated with dynein storage buffer alone instead of dynein motors before the addition of 30 μM LIS1 and microtubules as described in the Materials and methods. Video represents 2 min of real time; width of each frame is 80 μm.

unexpected functions of LIS1 in regulating the behaviour of dynein complexes (summarised in *Figure 9*).

## Insights into plus-end tracking of dynein complexes and the relationship to cargo transport

The targeting of mammalian dynein complexes to the plus ends of dynamic microtubules was previously investigated in vitro with a truncated version of a tissue-specific p150 isoform, which has the unusual property amongst p150 variants of only binding to microtubules in the presence of EB1 (*Dixit et al., 2008*; *Duellberg et al., 2014*). It was found that this p150 fragment induces robust targeting of dynein complexes to the plus end of growing microtubules by providing a link between EB1 and the dynein tail (*Duellberg et al., 2014*). However, this study did not discriminate between maintenance of dynein at the growing plus end through rapid turnover of motor complexes, or through persistent individual binding events. We have analysed single binding events of dynein at the plus end of growing microtubules in the presence of EB1 and the native brain dynactin complex and found a mean duration of tracking events of ~4 s, with addition of LIS1 increasing this value to ~14 s.

These dwell times are much longer than those reported for individual EB1 molecules at the growing plus end, which rarely exceed 1 s (*Bieling et al., 2008*; *Dixit et al., 2009*). In the filamentous fungus *Ustilago maydis*, EB1- and dynactin-dependent association of dynein with plus ends at the hyphal tip was also characterised by a slow turnover of individual motor complexes (*Schuster et al., 2011*). Our data suggest that persistent association of dyneins with dynamic microtubule plus ends could be an intrinsic property of the EB1/dynactin/dynein system. It remains to be determined whether dynein remains bound to the polymerising plus end through dynamic interactions with exchanging EB1 molecules, or whether EB1 is only required for the initial recruitment of the motor complex to this site.

Our study also provides insight into the contribution of other proteins to the association of mammalian dynein with EB1-bound plus ends. Whereas dynactin is essential for tracking of dynein at these sites, BICD2N is not. This observation suggests that dynein and dynactin associate with each other during tracking events in the absence of BICD2N. It was previously reported that in solution (*McKenney et al., 2014*; *Schlager et al., 2014*; *Splinter et al., 2012*), as well as on stabilised microtubules (*McKenney et al., 2014*), the presence of an adaptor protein such as BICD2N is required for formation of a stable complex between dynein and dynactin. Our data suggest that the formation of the dynein-dynactin complex is possible at EB1-bound microtubule plus ends in the absence of a cargo adaptor. Strikingly, in conjunction with EB1 and dynactin, LIS1 strongly increases the duration of binding events of human dynein at the growing plus end without affecting the duration of binding to lattice sites. This could conceivably be related to LIS1's ability to promote the association of dynein with the dynactin complex (*Dix et al., 2013*; *Wang et al., 2013*), which is expected to be enriched at the plus ends of growing microtubules through its interaction with EB1 (*Honnappa et al., 2006*). Our finding that LIS1 can promote the assembly of the dynein-dynactin-BICD2N complex in vitro is consistent with a role in stimulating the association of dynein with dynactin.

Our results also shed light on the influence of proteins that bind the growing plus end on the behaviour of minus end-directed transport complexes. EB1 made minus end-directed movements of dynein-dynactin-BICD2N complexes six times more likely to initiate at the plus end than a GDP

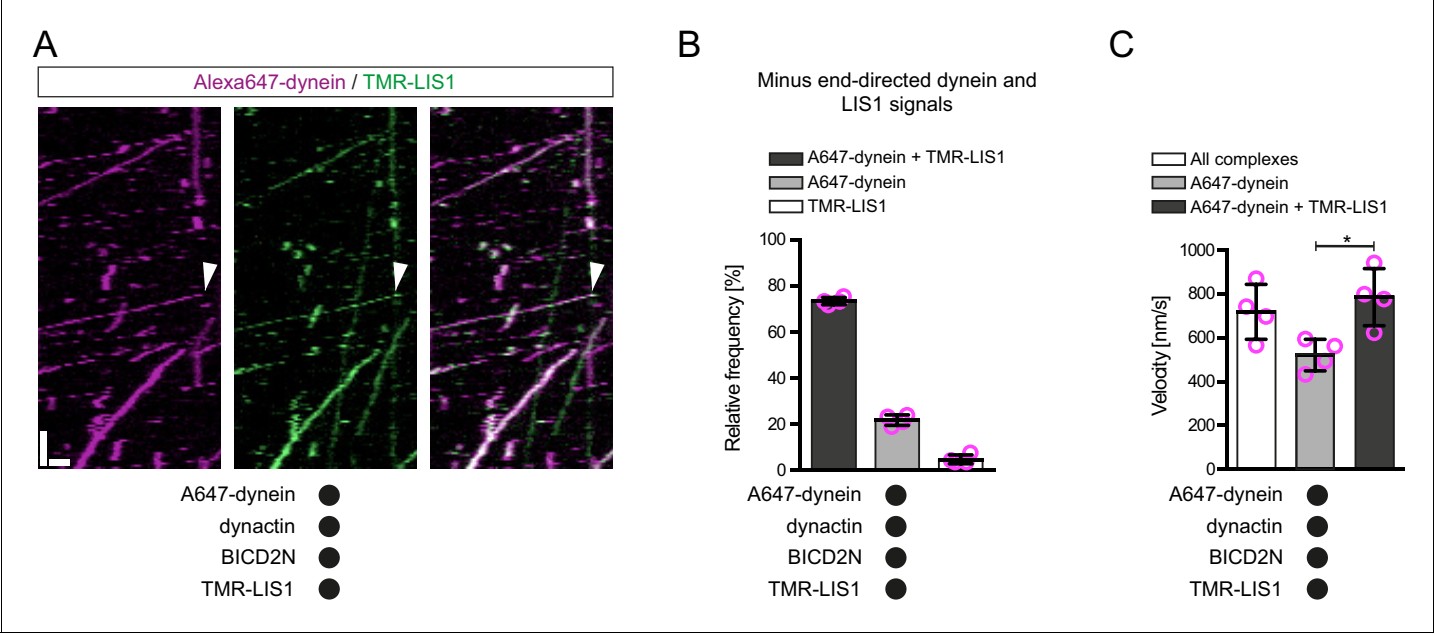

**Figure 7.** LIS1 frequently associates with dynein complexes undergoing minus end-directed transport. (**A**) Kymograph from a dynamic microtubule showing localisation of TMR-LIS1 on minus end-directed Alexa647-dynein complexes (e.g. arrowhead) in the presence of dynactin and BICD2N. Y-axis, time; x-axis, distance. Scale bars, 10 s and 1 μm. (**B**) Quantification of the proportion of minus end-directed transport events of Alexa647-dynein or TMR-LIS1 on dynamic microtubules with detectable signals from both proteins. (**C**) Quantification of the velocity of minus end-directed movements along dynamic microtubules that have detectable Alexa647-dynein signal alone, or signals from both Alexa647-dynein and TMR-LIS1. Minus end-directed complexes labelled with only TMR-LIS1, in which the dynein must be unlabelled or photobleached, were too few in number for meaningful velocity analysis. In B and C, means ± S.D. are shown with values for each chamber represented as magenta circles (four chambers per condition, with a mean of 104 and 110 complexes analysed per chamber in B and C, respectively. In C, statistical significance was evaluated with a Mann-Whitney test (*p<0.05). Dynactin complexes, BICD2N dimers and LIS1 dimers were used, respectively, at a molar excess of 2x, 10x and 20x compared to dynein. Dynein concentration in the assembly mixes was 20 nM, with a 1 in 5 dilution added to the imaging chambers.

The following figure supplement is available for figure 7:

**Figure supplement 1.** Kymograph exemplifying co-localisation of TMR-LIS1 with end tracking and static Alexa647-dynein in the presence of EB1, dynactin and BICD2N.

lattice site of the same length. The ability of EB proteins to bias the initiation of processive transport to the plus ends of microtubules could play a particularly important role in axons, where is has been proposed that the initiation of motility in distal regions is a rate-limiting step in retrograde transport processes (*Moughamian and Holzbaur, 2012*; *Moughamian et al., 2013*). Consistent with this notion, mutations in the CAP-Gly domain of p150 – which associates with EB1 – compromise the initiation of retrograde motility in distal axons and are associated with human neurological disease (*Farrer et al., 2009*; *Lloyd et al., 2012*; *Moughamian and Holzbaur, 2012*).

It has been hypothesised that tracking of dynein on plus ends of growing microtubules serves to efficiently capture cargoes destined for minus end-directed transport (*Lenz et al., 2006*; *Lloyd et al., 2012*; *Vaughan, 2004*; *Vaughan et al., 1999*, *2002*). We rarely observed plus-end tracking events of dynein being converted into minus end-directed transport events of dynein-dynactin-BICD2N complexes. Instead, the initiation of minus end-directed transport from the growing plus end was almost always co-incident with initial engagement of the motor complex with this site. Factors not included in our in vitro experiments may be able to promote the conversion of plus-end tracking events to minus end-directed transport events in vivo. Candidate factors include proteins that link cargoes to BICD2, such as Rab6 and other adaptors (*Hoogenraad and Akhmanova, 2016*). Post-translational modifications of tubulin, which have emerged as important regulators of dynein behaviour (*McKenney et al., 2016*; *Nirschl et al., 2016*), could also play a role in activation of minus end-directed movement from dynein-tracking events. Alternatively, tracking of dynein on the plus

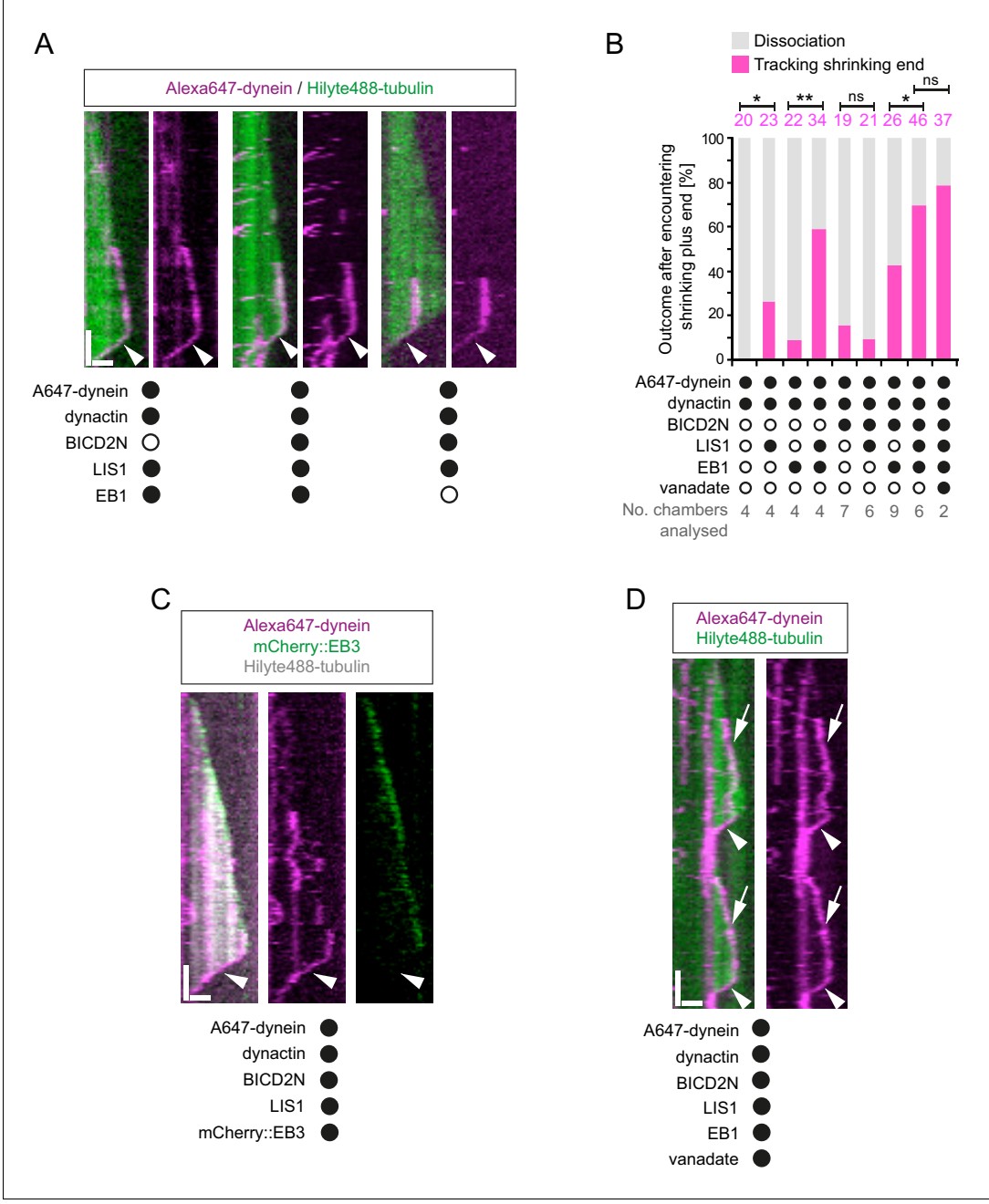

**Figure 8.** Characterisation of dynein association with plus ends during shrinkage phases. (**A**) Kymographs showing examples of dynein retention on plus ends undergoing shrinkage (arrowheads). In **A**, **C** and **D**: y-axis, time; x-axis, distance; scale bars, 10 s and 1 µm. (**B**) Quantification of outcome of microtubule-associated dyneins encountering the plus end of a shrinking microtubule. Fifteen microtubules were scored per chamber; magenta numbers above bars indicate the number of scored encounters of dynein with a shrinking end. Statistical significance was evaluated with a Fisher's exact test (**p<0.01; *p<0.05; ns, not significant). (**C**) Kymograph showing that mCherry:: EB3 is not present on shrinking plus ends bound by dynein (arrowhead). (**D**) Kymograph showing that maintenance of dynein on the plus end of microtubules during shrinkage (arrowheads) and growth phases (arrows) is not abolished by ATP.vanadate. Dynactin complexes, BICD2N dimers and LIS1 dimers were used, respectively, at a molar excess of 2x, 10x and 20x compared to dynein. Dynein concentration in the assembly mixes was 20 nM, with a 1 in 10 dilution added to the imaging chambers.

The following figure supplement is available for figure 8:

**Figure supplement 1.** Additional data relevant to the association of dynein with plus ends undergoing shrinkage.

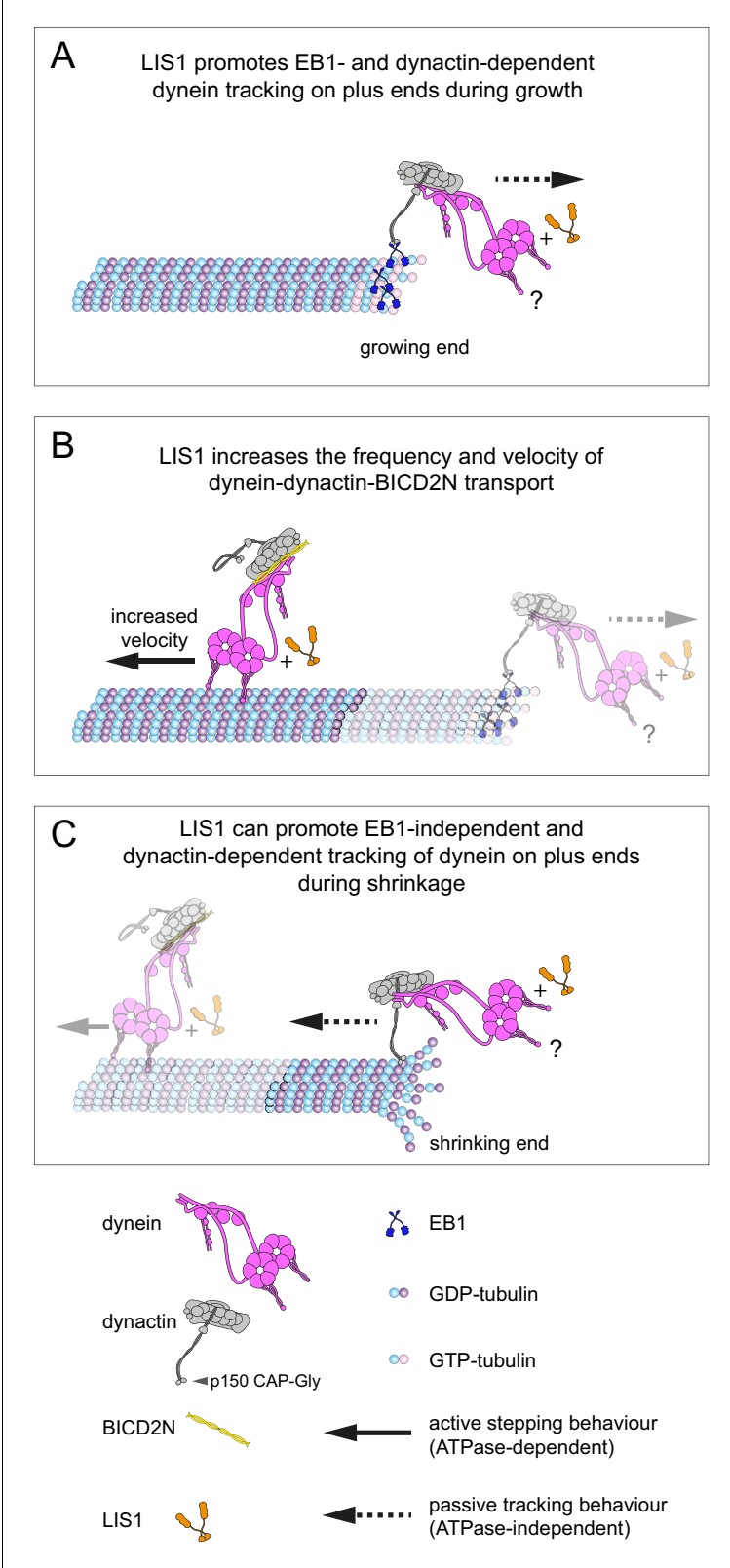

**Figure 9.** Model for the roles of LIS1 in regulating dynein behaviour on dynamic microtubules. (**A**) LIS1 promotes EB1- and dynactin-dependent association of dynein with growing microtubule plus ends by increasing the frequency and duration of tracking events. Although BICD2N is not required for end tracking behaviour of dynein, it can associate with tracking complexes. (**B**) LIS1 increases the frequency of minus end-directed movements of

*Figure 9 continued on next page*

*Figure 9 continued*

dynein in the presence of dynactin and BICD2N. LIS1 also increases the velocity of minus end-directed dynein-dynactin-BICD2N complexes by associating with them. (**C**) The presence of LIS1 can increase the likelihood of dynein remaining associated with the microtubule upon encountering a shrinking plus end. (**A, C**) The tracking of dynein on both growing and shrinking microtubule plus ends appears to be mediated by indirect coupling of the dynein tail to these sites by dynactin. During growth phases, coupling of dynein to the plus end involves microtubule binding by EB1, which in turn uses its EEY/F and EB homology motifs to recruit the CAP-Gly of p150 (**Honnappa et al., 2006**). During shrinkage, the coupling is independent of EB1 and presumably involves the microtubule binding activity of dynactin, which also involves the CAP-Gly domain of p150. The microtubule-binding domains of dynein could conceivably stabilise the dynactin-dependent association of dynein with growing and shrinking plus ends (**?**). Note that LIS1 has not been placed in a specific position on the dynein-dynactin-BICD2N complex as previous studies have provided evidence for more than one binding site (see Discussion).

ends of microtubules could have a role other than determining the initiation site of cargo transport. We did not detect changes in microtubule growth rates co-incident with binding and unbinding of dynein complexes. Thus, we have no evidence that plus-end tracking behaviour of dynein serves to directly regulate microtubule dynamics. Another possibility is that tracking of a subset of dynein complexes on dynamic microtubules helps deliver the motor to other sites in the cell (**Mimori-Kiyosue and Tsukita, 2003**; **Tamura and Draviam, 2012**).

## Mammalian LIS1 enhances the frequency and velocity of minus end-directed dynein movements

We found that LIS1 significantly increases the proportion of microtubule-associated dynein complexes that undergo minus end-directed transport in the presence of BICD2N and dynactin, without influencing the site of transport initiation on growing microtubules bound by EB1. Our data indicate that stimulation of minus end-directed transport by LIS1 is associated with an ability to promote the assembly of the dynein-dynactin-BICD2N complex. Consistent with an in vivo role for LIS1 in promoting processive movement of dynein-dynactin-cargo adaptor complexes, knockdown of LIS1 in mammalian cells strongly inhibits the pericentrosomal relocalisation of cargoes that is induced upon artificial tethering to BICD2N (**Splinter et al., 2012**).

We also found that LIS1 associates with dynein-dynactin-BICD2N complexes during minus end-directed motion and increases their velocity. The changes in speed induced by LIS1 are sizeable, with mean velocity increases in the range of 60–175% depending on the experimental conditions. LIS1 may therefore contribute to the high velocities exhibited by dynein-associated cargoes in vivo, which often exceed those of minimal motor complexes in vitro. For instance, several dynein-bound cargoes in mammalian cells have a mean velocity of 0.6–2 µm/s (**Flores-Rodriguez et al., 2011**; **Hafezparast et al., 2003**; **Ori-McKenney et al., 2010**; **Rai et al., 2013**), whereas in vitro movements of dynein-dynactin-BICD2N complexes in the absence of LIS1 have a mean velocity of 0.3–0.6 µm/s ([**Hoang et al., 2017**; **McKenney et al., 2014**, **2016**; **Olenick et al., 2016**; **Schlager et al., 2014**]; this study). The ability of LIS1 to stimulate dynein velocity is all the more remarkable because several in vitro studies have shown that LIS1 inhibits the activity of dynein complexes in the absence of other co-factors. It was reported that LIS1 reduces the velocity of microtubule gliding by surface immobilised yeast (**Huang et al., 2012**), pig (**Torisawa et al., 2011**; **Yamada et al., 2008**) or bovine (**Wang et al., 2013**) dynein, and we found that this is also the case in gliding assays with the human motor complex. LIS1's inhibition of microtubule translocation by dynein is presumably related to its ability to increase the interaction between the motor complex and the microtubule (**Huang et al., 2012**). In the case of mammalian dynein, LIS1 may exert this effect by prolonging the interaction of dynein with microtubules under load (**McKenney et al., 2010**), increasing the association rate of dynein for microtubules when in a large molar excess (this study), or a combination of these mechanisms. Intriguingly, the velocity of single yeast dynein complexes, which move processively without dynactin or a cargo adaptor (**Reck-Peterson et al., 2006**), was also reduced by increasing the concentration of LIS1 (**Huang et al., 2012**), as was that of isolated mammalian dynein complexes attached to beads (**McKenney et al., 2010**). Our data do not contradict these findings, which were made with significantly different experimental systems. However, they do paint a more complex

picture of LIS1 function, in which it has very different effects on dynein behaviour in the presence and absence of activators of processive minus end-directed motion.

How might LIS1 stimulate dynein velocity in the context of processive dynein-dynactin-BICD2N complexes? There is a consensus in the literature that LIS1 does not significantly increase the ATPase activity of dynein (*Huang et al., 2012*; *McKenney et al., 2010*; *Toropova et al., 2014*; *Yamada et al., 2008*; *Zhang et al., 2010*). Thus, the large velocity increase induced by LIS1 is unlikely to be due to a general stimulation of the motor's ATPase rate. It is, however, conceivable that LIS1 stimulates ATPase activity of dynein specifically when the motor complex is associated with dynactin and BICD2N. Alternatively, LIS1 could boost translocation rates without changing the kinetics of the ATP cycle, for example by increasing the step size of dynein within the dynein-dynactin-BICD2N complex.

## Tracking of dynein on plus ends during microtubule shrinkage

We observed that, in the presence of dynactin, motor complexes can remain attached to the microtubule plus end during shrinkage phases. In another example of its ability to regulate association of dynein on a specific site of the microtubule, LIS1 increased the probability of microtubule-associated motor complexes retreating with the shrinking end in several contexts. Dynein can associate with both growing and shrinking microtubule ends in budding yeast and filamentous fungi (*Han et al., 2001*; *Sheeman et al., 2003*). Association of dynein with dynamic plus ends in these systems involves CLIP-170 and LIS1 orthologues, which can couple the motor complex to plus-end-directed kinesin motors (*Carvalho et al., 2004*; *Efimov et al., 2006*; *Roberts et al., 2014*; *Sheeman et al., 2003*; *Zhang et al., 2003*). Our in vitro experiments demonstrate that the mammalian dynein complex can associate with shrinking ends for prolonged periods without CLIP-170 and a plus-end-directed motor. What then is the underlying basis of this behaviour? We show that end tracking of the motor complex during microtubule shrinkage is not dependent on ATP hydrolysis or an EB protein, but is dependent on dynactin. These observations point towards a biased diffusion mechanism for retention of dynein, presumably involving the microtubule-binding activities of p150 (*Ayloo et al., 2014*; *Culver-Hanlon et al., 2006*; *Lazarus et al., 2013*; *Wang et al., 2014*; *Waterman-Storer et al., 1995*) and/or dynein. As is the case on growing plus ends, tracking of dynein on shrinking plus ends is not dependent on BICD2N (*Figure 8B*). This result again points towards context-dependent mechanisms for dynein-dynactin complex formation in the absence of a cargo adaptor.

Our in vitro studies of dynein tracking on shrinking ends pave the way for investigations into the relevance of this behaviour for dynein function in vivo. By prolonging the interaction with dynamic microtubules, an ability of mammalian dynein-dynactin complexes to track shrinking ends would be expected to increase the delivery of the motor complex to other cellular sites. Association of dynein with shrinking ends may also be beneficial in other contexts, such as when the cortically anchored dynein exerts a pulling force on depolymerising microtubules in mitotic or interphase cells (*Burakov et al., 2003*; *Dujardin and Vallee, 2002*; *Grill and Hyman, 2005*; *Laan et al., 2012*; *Nguyen-Ngoc et al., 2007*; *Ten Hoopen et al., 2012*).

## Perspective

Much of our thinking about LIS1 function has been influenced by in vitro studies in the presence of isolated dynein complexes and artificially stabilised microtubules. Our experiments in the presence of dynamic microtubules and other important dynein regulators have provided additional insights into the function of LIS1 in mammalian systems (summarised in *Figure 9*). Our finding that LIS1 can stimulate the frequency and velocity of processive movement of a dynein-dynactin-cargo adaptor complex offers another explanation for the basis of dynein-activating functions of LIS1 that are often observed in vivo. It has also been shown recently in mammalian cells that LIS1, but not dynactin, contributes to the ability of dynein on a cellular cargo to adapt to forces exerted by an optical trap (*Reddy et al., 2016*). A key question for the future is how LIS1 exerts its context- and concentration-dependent effects on dynein complexes. Several publications have demonstrated that LIS1 can bind to the isolated motor domain of dynein (*Huang et al., 2012*; *McKenney et al., 2010*; *Sasaki et al., 2000*; *Tai et al., 2002*), with a structure of the yeast proteins revealing association of LIS1 with the junction of the dynein AAA3 and AAA4 domains in proximity to the linker element (*Huang et al.,*

*2012*; *Toropova et al., 2014*). Differential effects of LIS1 could be related to occupancy of this binding site on one versus two of the dynein heads. Another possibility is suggested by earlier yeast two hybrid and pulldown experiments examining the interaction between LIS1 and mammalian dynein (*Sasaki et al., 2000*; *Tai et al., 2002*). This work provided evidence that LIS1 can contact the AAA1 domain and tail region of dynein heavy chain, as well as dynein intermediate chain and the p50 subunit of the dynactin complex (*Sasaki et al., 2000*; *Tai et al., 2002*). We therefore speculate that interactions of LIS1 with different parts of the dynein complex contribute to the contrasting regulatory effects observed in this and other studies. Future experiments should be aimed at understanding the structural basis of LIS1's regulation of dynein behaviour in different contexts. It will also be important in the future to build further complexity into in vitro studies of dynein's interactions with dynamic microtubules. It will be particularly interesting to assess the function of NudE and NudEL in our in vitro reconstitution system, as these proteins can strongly influence the interactions of dynein with both dynactin and LIS1 (*McKenney et al., 2010*, *2011*; *Wang et al., 2013*).

## Materials and methods

### Cloning and protein production

The human dynein complex (which consists of six subunits, each present in two copies per complex) was expressed in Sf9 cells (from *Spodoptera frugiperda*; ThermoFisher Scientific (Waltham, MA) (Cat. No. 11496015)) and purified as described previously (*Schlager et al., 2014*). The cells have not recently been genetically profiled or evaluated for contamination with mycoplasma. However, this is not relevant to the findings of the study, which only uses recombinant proteins purified from these cells. Cells were infected with a baculovirus containing an integration of the *pDyn3* plasmid. This plasmid contains the following sequences codon optimised for Sf9 cell expression and placed downstream of polyhedrin promoters: DHC (DYNC1H1, accession number NM_001376.4), DIC2 (DYNC1I2, IC2C, AF134477), DLIC2 (DYNC1LI2, LIC2, NM_006141.2), Tctex (DYNLT1, Tctex1, NM_006519.2), LC8 (DYNLL1, LC8-1, NM_003746.2) and Robl (DYNLRB1, Robl1, NM_014183.3). The DHC gene was fused at its 5' to sequences encoding a His-ZZ-LTLT tag (*Reck-Peterson et al., 2006*). The ZZ tag is a tandem IgG binding domain based on protein A of *Staphylococcus aureus* and the LTLT region includes a cleavage site for the tobacco etch virus (TEV) protease. The presence of each gene in the expression construct was verified by PCR using Quickload Taq 2x Master Mix (New England Biolabs (NEB; Ipswich, MA)). The ZZ tag on DHC was used for affinity purification of the dynein complex with IgG Sepharose 6 FastFlow beads (GE Healthcare (Chicago, IL)) prior to a TEV protease-mediated elution step.

Human EB1 was expressed in *E. coli* using an expression plasmid gifted by T. Mitchison and J. Tirnauer (Plasmid #39297 from Addgene (RRID:SCR_002037; Cambridge, MA)). EB1 was purified via a His tag as described (*Tirnauer et al., 2002*). Recombinant human mCherry::EB3 was expressed in *E. coli* and purified via a His tag as described (*Montenegro Gouveia et al., 2010*). Recombinant human GFP::CLIP-170 and CLIP-170 were expressed in Sf9 cells and purified via a His tag as described (*Bieling et al., 2008*). The mCherry::EB3, GFP::CLIP-170 and CLIP-170 expression constructs were kindly provided by T. Surrey (Crick Institute, London). SNAP$_f$::BICD2N was expressed in Sf9 cells. The SNAP$_f$::BICD2N expression construct (*Belyy et al., 2016*; gift of A. Carter, MRC-LMB) contains Sf9 codon-optimised sequences producing residues 1–400 of mouse BICD2 (which are 95% identical to the equivalent BICD2N residues in human) fused to N-terminal ZZ-His-LTLT-SNAP$_f$ sequences. SNAP$_f$::BICD2N was purified using IgG Sepharose 6 Fast Flow beads using the protocol established for GFP::BICD2N (*Schlager et al., 2014*). Large-scale SP-sepharose-based purification of native dynactin from pig brain was performed as described (*Urnavicius et al., 2015*).

The pFastbac expression construct for human full-length LIS1 with an N-terminal His-ZZ-LTLT tag was kindly provided by A. Carter (MRC-LMB). To produce the LIS1::SNAP$_f$ expression construct, sequences encoding a Gly-Ala-Gly-Ala-Gly-Ala linker followed by a SNAP$_f$ tag were inserted between the C-terminal residue of LIS1 and the SV40 termination sequences of the ZZ-His-LTLT-LIS1 plasmid using Gibson Assembly Master Mix (NEB). All purification steps for LIS1 and LIS1::SNAP$_f$ were performed at 4°C. Frozen pellets from 50 ml of Sf9 cells were thawed on ice and resuspended in 2 ml of Lysis Buffer (50 mM Tris HCl pH 8, 250 mM KOAc, 2 mM Mg(OAc)$_2$, 1 mM EGTA, 10% (v/v) glycerol, 1 mM DTT, 2 mM PMSF). Cells were lysed with 15 strokes of a 1 ml dounce homogenizer

(Wheaton, UK). The lysate was cleared of cell debris by centrifugation at 4°C in an Eppendorf 5424 microfuge (20,238 x *g* for 15 min). The cleared lysate was incubated in 2-ml tubes with 400 µl IgG Sepharose 6 Fast Flow beads that had been pre-washed in Lysis Buffer. The lysate and beads were incubated for 2 hr at 4°C on a flat roller, followed by transfer into a 5 ml gravity flow polypropylene column (Qiagen (Hilden, Germany)). The beads were washed three times with 5 ml TEV Buffer (Lysis Buffer without PMSF) and resuspended in 1 ml of TEV buffer, followed by transfer to a 1.5-ml tube. The beads were pelleted by centrifugation (30 s, 94 x *g*), resuspended in 300 µl TEV Buffer and transferred to a 0.5-ml tube. To fluorescently label the SNAP$_f$ tag on LIS1, the LIS1::SNAP$_f$ bound beads were incubated with SNAP-Cell TMR-Star (NEB) at this stage as described below. To elute LIS1 or LIS1::SNAP$_f$ from the beads, 25 µl of 4 mg/ml TEV protease was added and the tube filled with TEV Buffer. Tubes were incubated on a tube roller overnight at 4°C. The beads were then pelleted by centrifugation (30 s, 94 x *g*) and the supernatant used for size exclusion chromatography on a Superdex 200 Increase column on an ÄKTAmicro system (GE Healthcare) in TEV buffer. The fractions containing LIS1 were pooled, followed by dispensing of small aliquots and flash freezing.

SDS–PAGE of purified proteins was performed using Novex 4–12% Bis–Tris precast gels using either MOPS or MES buffer (Life Technologies (Carlsbad, CA)) and ECL Full-Range Rainbow Molecular Weight Markers (GE Healthcare). Gels were stained with the Coomassie-based reagent Instant Blue (Expedeon (San Diego, CA)) or Imperial Protein Stain (ThermoFisher Scientific) according to the manufacturers' instructions and imaged using a ChemiDoc XRS+ system with Image Lab 4.0 software (Bio-Rad (Hercules, CA)). Protein concentrations were measured using the Coomassie Protein Assay Kit (ThermoFisher Scientific) and an Eppendorf BioPhotometer Plus (Eppendorf (Hamburg, Germany)).

## Labelling of SNAP proteins

IgG Sepharose 6 Fast Flow beads bound to SNAP$_f$::dynein complexes or LIS1::SNAP$_f$ were incubated at 4°C for 40 min with ~5 µM SNAP-Surface Alexa Fluor 647 or SNAP-Cell TMR-Star (NEB) for SNAP$_f$::dynein, or ~40 µM SNAP-Cell TMR-Star for LIS1::SNAP$_f$. Prior to TEV cleavage, excess dye was removed with three washes (25 ml each for dynein; 2 ml each for LIS1) in TEV Buffer, and beads resuspended in 300 µl TEV buffer. The labelling efficiency of dynein preparations was determined by spectrophotometric analysis of concentrated protein samples with a Nanodrop 1000 spectrophotometer (Nanodrop Technologies (Wilmington, DE)). The labelling efficiency was between 81% and 94% per dynein monomer, which equates to 96% to 99.6% of dimeric dynein complexes being labelled. The labelling efficiency of LIS1::SNAP$_f$ could not be evaluated by spectrophotometry as the protein could not be concentrated sufficiently without precipitating out of solution. However, the results of our co-localisation analysis of LIS1::SNAP$_f$ with dynein on microtubules (*Figure 2—figure supplement 3* and *Figure 7B*) indicates very efficient LIS1 labelling. Inefficient labelling of SNAP$_f$:: BICD2N was observed with the method described above, possibly because the dye was not in sufficient molar excess to the protein. SNAP$_f$::BICD2N was therefore labelled following TEV cleavage as described (*Hoang et al., 2017*), which allowed the concentration of the protein to be first determined accurately. This procedure resulted in a labelling efficiency of 80% per SNAP$_f$::BICD2N monomer, equating to 96% of the dimers being labelled.

## TIRF microscopy

Imaging was performed at 24 ± 1°C. The majority of data were acquired on a TIRF microscope system (Nikon (Amsterdam, Netherlands)) controlled with Micro-Manager software (*Edelstein et al., 2010*; *Edelstein et al., 2014*; RRID:SCR_000415). The microscope was equipped with a 100× objective (Nikon, 1.49 NA Oil, APO TIRF) and the following lasers: 150 mW 488 nm, 150 mW 561 nm laser (both Coherent Sapphire (Coherent Inc. (Santa Clara, CA)), and 100 mW 641 nm (Coherent CUBE (Coherent Inc.)). Images were acquired with an EMCCD camera (iXon$^{EM}$+ DU-897E, Andor (Belfast, UK)). For multicolour experiments, images were captured sequentially by switching emission filters between GFP, Cy3, and Cy5 (Chroma Technology Corp. (Bellows Falls, VT)). The size of each pixel was 105 nm × 105 nm. The data in *Figure 3*, *Figure 3—figure supplement 1* and *Figure 6—figure supplement 2B* were collected with a Nikon TIRF system controlled with Nikon Elements v4.3. The microscope was equipped with the same type of 100x objective and camera as described above. 80 mW 488 nm and 125 mW 647 nm lasers (Agilent (Santa Clara, CA)) and ET525/

50m and ET705/72m emission filters (Chroma Technology Corp.) were used. The size of each pixel was 160 nm × 160 nm.

## Flow chamber preparation

PEG and Biotin-PEG-functionalised glass surfaces and PLL-PEG passivated counter glass surfaces were prepared as described previously (*Bieling et al., 2010*). PEG and Biotin-PEG were purchased from Rapp Polymere (Tuebingen, Germany), and PLL-PEG was purchased from SuSos AG (Duebendorf, Switzerland). Flow chambers were constructed as described previously (*Bieling et al., 2010*).

## Production of GMPCPP-stabilised microtubules

Unlabelled porcine tubulin, Hilyte488-tubulin and biotin-tubulin were supplied by Cytoskeleton Inc. (Denver, CO). To polymerise fluorescent GMPCPP-stabilised microtubules, 1.66 µM unlabelled tubulin, 0.15 µM Hilyte488-tubulin and 0.4 µM biotin-tubulin were incubated together with 0.5 mM GMPCPP (Jena Bioscience (Jena, Germany)) in BRB80 (80 mM PIPES pH 6.85, 2 mM $MgCl_2$, 0.5 mM EGTA) for 2 hr 30 min at 37°C. Polymerised microtubules were sedimented by centrifugation at 18,400 x *g* in an Eppendorf 5424 centrifuge for 8 min at room temperature. The microtubule pellet was washed by resuspension in 1 ml of prewarmed BRB80 and sedimented again by centrifugation. Microtubules were then resuspended in BRB80 that had been prewarmed to 37°C and kept at room temperature for use within 1–2 hr.

## Assaying dynein behaviour on dynamic and stabilised microtubules

Flow chambers were prepared as described above and passivated for 5 min with 1% (w/v) pluronic F-127 (Sigma (St Louis, MO)) in $dH_2O$. Chambers were washed twice with BRB80 and treated with 2 mg/ml streptavidin (Sigma) for 5 min, followed by two washes with BRB80. Exposed glass surfaces were then blocked with 5 mg/ml α-casein (Sigma) dissolved in BRB80 for 5 min. Fluorescent, biotiny-lated GMPCPP-stabilised microtubule seeds (polymerised as described above) were immobilised on the biotin-PEG-coated glass surface via streptavidin as described previously (*Bieling et al., 2010*). Chambers were washed once with Assay Buffer (80 mM PIPES pH 6.85, 2 mM $MgCl_2$, 0.5 mM EGTA, 0.5 mg/ml BSA, 2.5 mM ATP, 1 mM DTT). For each imaging chamber, dynein was incubated for 5 min on ice with combinations of dynactin, BICD2N, CLIP-170 and LIS1 in Assay Buffer ('Assembly Mix'). The concentration of dynein dimers in assembly mixes was 20 nM. In experiments evaluating the effects of LIS1, control protein mixes lacking LIS1 were supplemented with LIS1 storage buffer to ensure identical buffer conditions. Dynactin (1.1 MDa per complex), $SNAP_f$::BICD2N dimers (134 kDa per complex), LIS1 dimers (93 kDa per complex without the $SNAP_f$ tag, and 132 kDa per complex with the $SNAP_f$ tag) and CLIP170 dimers (324 kDa per complex) were used in, respectively, a 2-fold, 10-fold, 20-fold and 25-fold molar excess to the full dynein complex (1.42 MDa per complex). Assembly mixes were diluted 1 in 2 to 1 in 10 to give a 'chamber mix' solution with a final concentration of 100 nM EB1 or mCherry::EB3 dimers, 26 µM unlabelled tubulin, 1.12 µM Hilyte488-tubulin, 2.5 mM ATP, 2 mM GTP, 0.2% (w/w) methyl cellulose, 10 mM KOAc, 10 mM KCl, 1250 nM glucose oxidase, 140 nM catalase, 71 mM 2-mercaptoethanol, 24.9 mM glucose, 80 mM PIPES pH 6.85, 2 mM $MgCl_2$, 0.5 mM EGTA, 0.5 mg/ml BSA and 1 mM DTT (see legends for information on dilution factors for specific experimental series). Varying the dilution factor for the assembly mixes was necessary at certain points in the study to give densities of complexes on microtubules that were conducive to the analysis of single binding events. The concentrations of dynein in the assembly mix and chamber mix were kept constant in any experiments that were compared to each other. In a subset of experiments, sodium orthovanadate (vanadate (NEB)) was added to a final concentration of 100 µM (i.e. 2.5 mM ATP +100 µM vanadate). In the experiment documented in *Figure 2—figure supplement 1*, a 500 nM solution of GFP::CLIP170 was diluted 1:10 to give a solution with a final concentration of 100 nM EB1 dimers, 26 µM unlabelled tubulin, 1.12 µM Hilyte488-tubulin, 2 mM GTP, 0.2% (w/w) methyl cellulose, 10 mM KOAc, 10 mM KCl, 1250 nM glucose oxidase, 140 nM catalase, 71 mM 2-mercaptoethanol, 24.9 mM glucose, 80 mM PIPES pH 6.85, 2 mM $MgCl_2$, 0.5 mM EGTA, 0.5 mg/ml BSA and 1 mM DTT (i.e. as above but without ATP).

Assessment of the behaviour of dynein or dynein-dynactin-BICD2N complexes on non-dynamic microtubules was assayed as described above, with two differences. First, microtubules were

resuspended in BRB80 supplemented with 40 µM taxol (paclitaxel (Sigma)). Second, the final 'chamber mix' contained a dilution of the assembly mix in 2.5 mM ATP, 10 mM KOAc, 10 mM KCl, 1250 nM glucose oxidase, 140 nM catalase, 71 mM 2-mercaptoethanol, 24.9 mM glucose, 80 mM PIPES pH 6.85, 2 mM $MgCl_2$, 0.5 mM EGTA, 0.5 mg/ml BSA and 1 mM DTT. The dynein concentration in the assembly mix for these experiments was also 20 nM, except for *Figure 3* and *Figure 5B–D* when it was 100 nM (see legends for experimental details).

Chamber mixes were introduced into flow chambers, which were subsequently sealed with nail polish. Flow chambers were then visualised by TIRF microscopy. Acquisition parameters were typically as follows: 100 ms exposures for one-colour imaging with a total acquisition time per frame of 0.137 s; 100 ms exposures per channel for two-colour imaging and a total acquisition time for both channels of 0.6 s; 50 ms exposure per channel for three-colour imaging and a total acquisition time for all three channels of 0.66 s. The exception was for the experiment in *Figure 3*, when the exposure time and total acquisition times were 100 ms and 0.5 s/frame, respectively. In experiments with stabilised microtubules, the positions of microtubules were recorded by capturing a single image of the fluorescent tubulin signal before acquiring a time series in the other channel(s). For each chamber, three movies (each of a different region of the chamber) were recorded, each lasting ~5 min. Kymographs of dynein complexes on five randomly selected microtubules per movie were generated with FIJI software (*Schindelin et al., 2012*; RRID:SCR_002285) and analysed manually.

In dynamic microtubule assays, the microtubule plus end was identified by its substantially faster growth rate compared to the minus end (*Summers and Kirschner, 1979*). We classified dynein-binding events on growing microtubule ends as 'tracking' if they overlapped with the trajectory of the growing microtubule tip for three or more frames (≥1.8 s). For consistency, events that lasted <1.8 s were also excluded from the analysis of the duration of dynein binding events on the microtubule lattice, when categorising dynein complexes as static, diffusive and processive in the presence of dynactin and BICD2N, and when counting landing events in *Figure 3*. For run length analysis, a run was defined as a bout of unidirectional, minus end-directed motion that could be terminated by either detachment from the microtubule or a pause (including at the minus end of the microtubule). Because a small subset of dynein complexes switched velocities during a run, mean velocities were calculated from individual constant velocity segments, as described previously (*Schlager et al., 2014*). The automatic tracking software FIESTA (*Ruhnow et al., 2011*; RRID:SCR_014990) was used to determine fluorescent intensity of minus end-directed Alexa647-dynein complexes on microtubules directly from the original image series. Fluorescence intensities of all unidirectional, minus end-directed dynein complexes were calculated as the volume under the Gaussian function. Velocities were also determined automatically with FIESTA for this analysis.

The following procedure was performed to determine the ratio of initial dynein docking events on the growing plus end versus an equivalently sized lattice site. A diagonal line ~0.1 pixels wide was drawn on a kymograph in FIJI to demarcate the plus end of the growing microtubule. Docking events at the plus end were then scored manually as the initial appearance of a dynein signal (regardless of the dwell time) that overlaps with the diagonal line. Docking events on the GDP lattice were scored with the same procedure using a parallel line of the same thickness that was positioned randomly (i.e., without previous visualisation of the dynein signal). The overlaps of dynein signals with diagonal lines drawn at growing plus end and randomly selected GDP lattice sites were also used to evaluate the initiation site of minus end-directed dynein movements.

Throughout the study, each chamber utilised a unique assembly mix. In experiments used for statistical evaluations different experimental conditions within one series were typically assayed in an interleaved manner on the same day with at least two days of experimentation in total. Data were analysed to confirm that there was no significant day-to-day variability between results.

## Microtubule gliding assays

Hilyte488-labelled microtubules were polymerised as described above, except that biotin-tubulin was not added to the polymerisation mix. Polymerised microtubules were resuspended in BRB80 containing 40 µM taxol that was prewarmed to 37°C. Flow chambers assembled from untreated glass surfaces were passivated for 5 min with 1% (w/v) pluronic F-127 in $dH_2O$, placed on an ice-cold metal block and washed with Assay Buffer. The following procedure was followed for all gliding assays except those documented in *Figure 6—figure supplement 3B* and *Video 2*. Proteins were diluted in Assay Buffer to give solutions containing 30 nM Alexa647-dynein alone, or 30 nM

Alexa647-dynein with 0.6 µM or 6 µM LIS1 dimers. When making dilutions, LIS1 storage buffer alone was used to ensure equivalent buffer conditions in all cases. Solutions of dynein or dynein plus LIS1 were incubated for 2 min on ice before introduction into the imaging chamber. Following a 5-min incubation on a metal block in ice, imaging chambers were allowed to warm up to room temperature by removal from the ice block for 2 min. Excess protein was then removed by one wash with 20 µl Assay Buffer. A solution (Microtubule Mix) containing a 1:10 dilution of Hilyte488-labelled microtubules in 2.5 mM ATP, 10 mM KOAc, 10 mM KCl, 1250 nM glucose oxidase, 140 nM catalase, 71 mM 2-mercaptoethanol, 24.9 mM glucose, 80 mM PIPES pH 6.85, 2 mM $MgCl_2$, 0.5 mM EGTA, 0.5 mg/ml BSA and 1 mM DTT was flowed into the chamber, which was promptly sealed with nail polish.

For the experiments documented in *Figure 6—figure supplement 3B*, LIS1 was added at a subsequent step to dynein in order to eliminate the risk that LIS1 interferes with microtubule gliding by competing with dynein for initial binding to the glass surface. The procedure above was followed except for the following modifications: glass surfaces were incubated with 300 nM Alexa647-dynein in Assay buffer for 5 min on ice and washed twice with Assay Buffer before blocking of any exposed surfaces with 20 mg/ml α-casein in 20 mM Tris, pH 8.0 for 5 min and a further two washes in Assay Buffer. 30 µM LIS1 dimers in Assay Buffer (or the equivalent volume of LIS1 storage buffer diluted in Assay Buffer) was then flowed into the chambers, followed by a 5 min incubation on ice. Following a subsequent incubation for 2 min at room temperature, the LIS1 solution was replaced with Microtubule Mix. An identical procedure was followed for the LIS1 only control documented in *Video 2*, except dynein storage buffer alone was used in the place of the dynein solution.

In all gliding assays, microtubules were visualised by TIRF microscopy for no longer that 15 min (during which time three regions of the chamber were imaged), with a 100 ms exposure and a frame interval of 1 s. The time between addition of microtubules and the start of imaging was kept constant (2 min) within each experimental series in order to control for any effects of gradual dissociation of LIS1 from dynein. However, we observed no decrease in the inhibitory effect of LIS1 on microtubule gliding by dynein during the period of image acquisition. This finding indicates that sufficient LIS1 associated with dynein for a strong inhibitory effect for the duration of the gliding assay.

Velocities of gliding microtubules were determined by manual analysis of kymographs with FIJI. In the gliding assays documented in *Figure 6—figure supplement 3B* the intensity of Alexa647-dynein on glass surfaces was also quantified. After collecting each time series of Hilyte488-microtubules, Alexa647-dynein signal was quickly captured from 10 to 14 randomly selected 80 µm x 80 µm regions of each imaging chamber. For each image, FIJI was used to measure the mean intensity of Alexa647 per pixel following background subtraction using a rolling ball with 50-pixel-radius.

## Statistics

Prism 6 (RRID:SCR_002798) (GraphPad (La Jolla, CA)) was used for data plotting and statistical analysis. Evaluations of statistical significance are described in the figure legends.

## Note in proof

Two preprints have recently been posted that also investigate the influence of LIS1 on mammalian dynein complexes in vitro (https://doi.org/10.1101/124255; https://doi.org/10.1101/126508)

## Acknowledgements

We are very grateful to L Urnavicius, H Foster, A Carter, V Madan (MRC-LMB) and members of the Bullock lab for generously providing advice and assistance during this project. We would also like to thank A Carter for feedback on the manuscript and T Surrey (Crick Institute, London) for providing the CLIP-170 and EB3 expression constructs. This work was supported by the UK Medical Research Council (file reference number MC_U105178790 (to SLB)), a Deutsche Forschungsgemeinschaft research fellowship (file reference number BA 5802/1–1 (to JB)) and a Boehringer Ingelheim Fonds Fellowship (to HH). RZ and AM thank A Carter and M Babu (MRC-LMB), respectively, for support during this project.

## Additional information

### Funding

| Funder | Grant reference number | Author |
|---|---|---|
| Medical Research Council | MC_U105178790 | Simon L Bullock |
| Deutsche Forschungsgemeinschaft | BA 5802/1-1 | Janina Baumbach |
| Boehringer Ingelheim Fonds | | Ha Thi Hoang |

The funders had no role in study design, data collection and interpretation, or the decision to submit the work for publication.

### Author contributions

JB, Conceived the project, Designed experiments, Performed the majority of reconstitution assays and analyses, Produced several proteins, Interpreted data, Wrote the manuscript; AM, MAM, Designed experiments, Performed a subset of reconstitution assays and analyses, Produced several proteins, Interpreted data, Edited the manuscript; CID, Produced several proteins and important insights, Edited the manuscript; RZ, Developed the purification protocol for LIS1 and provided initial LIS1 samples, Edited the manuscript; HTH, Developed the method for efficient fluorescent labelling of BICD2N and provided labelled protein, Edited the manuscript; SLB, Conceived the project, Designed experiments, Interpreted data, Wrote the manuscript, Supervised the project

### Author ORCIDs

Simon L Bullock, http://orcid.org/0000-0001-9491-4548

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
