## [Decision Letter]

Thank you for submitting your article "Novel Functions of Lissencephaly-1 Uncovered by Analysis of Mammalian Dynein Behaviour on Dynamic Microtubules" for consideration by *eLife*. Your article has been reviewed by 3 peer reviewers, and the evaluation has been overseen by Anna Akhmanova as the Senior and Reviewing Editor. The reviewers have opted to remain anonymous.

The reviewers have discussed the reviews with one another and the Reviewing Editor has drafted this decision to help you prepare a revised submission.

Summary:

In this work, Baumbach et al. investigate how the microtubule minus-end directed motor dynein accumulates at microtubule plus ends, a localization previously shown to be important for different dynein functions. By using in vitro reconstitution experiments, the authors systematically determine the requirements for dynein delivery to and tracking with the plus end of both growing and shrinking microtubules. Their experiments lead to several novel and surprising findings about dynein's interaction with microtubule plus ends. Key findings described here include: 1) While EB1 and dynactin are required for plus end targeting of dynein, the addition of LIS1 increases the frequency and duration of dynein's interactions with growing microtubule plus-ends. 2) Dynein-dynactin-BICD2N complexes, which are capable of long distance processive motility, are more likely to begin a run at the microtubule plus end in the presence of EB1, but EB1 does not alter the complexes velocity or run length. 3) Dynein-dynactin-BICD2N complexes that track with the growing microtubule plus end are rarely competent to engage in minus-end-directed motility. 4) LIS1 co-migrates with moving minus-end-directed dynein-dynactin-BICD2N complexes and increases the velocity of these complexes. This is a surprising finding as previous studies on LIS1 suggested that LIS1 decreases velocity. 5) LIS1, in conjunction with EB1 and dynactin, promotes dynein association with shrinking microtubules. In summary, this study defines novel ways in which Lis1 can impact on dynein motor behaviour.

Essential revisions:

1) The authors report a surprising ability of LIS1 to activate both the frequency and velocity of dynein-dependent runs. This observation appears to directly contrast with previous studies showing that LIS1 leads to dynein inhibition during gliding assays, and that the binding of LIS1 enhances the development of an ATPase insensitive state in which dynein remains bound to the microtubule. To the credit of the investigators, they repeat the gliding assay inhibition, which suggests that the difference is not due to differential activities of the purified dyneins used in the various labs. However, they provide no significant insights into the dramatic differences between their single molecule and gliding data, nor any insights into how their new observations can be reconciled with the many previous single molecule studies on LIS1 from the Reck-Peterson, Vallee, and Gross labs. Some explanation of the discrepancy between the single molecule and gliding assays should be provided. The authors should also provide much better controls and description for both the single molecule and gliding assays:

A) The authors should investigate whether Lis1 has an effect on human dynein microtubule interactions at the single molecule level, in the absence of other regulators (landing rate, dwell time, balance of static vs. diffusive events). The focus of this study is on more complex reconstitutions, but these data are fundamental and it seems very likely the authors already have them. Figure 2 compares dynein behaviour +/- Lis1 and dynactin, but the crucial +Lis1 kymograph seems to be missing from panel A.

B) A major experimental concern relates to the microtubule gliding assay (Figure 4), which the authors use to report that their Lis1 preparation reduces the velocity of microtubule gliding driven by human dynein. Apparently, dynein was non-specifically adsorbed to the flow chamber either alone or in the presence of 6 μm or 60 μm Lis1. Unbound proteins were then washed out. This approach raises two substantive concerns.

First, did the presence of Lis1 (up to ~2.7 mg/ml) reduce the amount of dynein that non-specifically adsorbed to the surface? Lis1 is a sticky protein, so this seems plausible, especially at the 200-fold Lis1 molar excess. The presence of microtubules on the surface does not rule this out, as microtubules adsorb to many surfaces, including those lacking motors. Either the consistency of dynein surface density in the different conditions needs to be confirmed (e.g. using fluorescent dynein) or, to avoid this danger entirely, the authors could specifically immobilize dynein, then subsequently add the desired concentration of Lis1.

Second, washing the chamber in a solution lacking Lis1 means that the dynein-Lis1 binding reaction will not have been at the desired equilibrium during the experiment: Lis1 will have been gradually dissociating in the time post wash out, and thus the fraction of dynein molecules with Lis1 bound will have depended critically on the time that elapsed between the washout and the start of the data acquisition. Was this elapsed time kept constant? It would be much safer to include Lis1 at the desired concentration in the final assay mixture.

2) The experiments with the dynein tail construct are somewhat confusing. Is the tail alone sufficient to track growing and shrinking plus ends? What is the minimum set of proteins required to allow the tail to track growing and shrinking ends? Can this tracking be seen with purified dynein, or is it limited to recombinant forms? Does Lis1 influence the frequency/duration of tracking events for the dynein tail, or is the motor domain required? (Alternatively, mutations that prevent Lis1 binding at the AAA4 site are known, and could be tested in the tracking assay.)

The authors propose the model that the different effects of Lis1 may arise from different binding sites, invoking interactions reported between Lis1 and various fragments of dynein and dynactin. However, interactions reported from the cited yeast 2-hybrid and pull-down experiments have not all been reproduced with intact proteins, so it is necessary to exercise caution before setting the field off down this path. Importantly, it seems that the authors may already have the data to test this model in their end-tracking experiments with the dynein tail construct. i.e. if Lis1 influences the frequency/duration of tracking events for the dynein tail, this would provide strong evidence for a functional binding site outside the characterised one in the dynein motor domain.

3) It is possible that dynein-dynactin-BICDN complexes that track the plus end contain a "dead" motor (Figure 3), or a dynein motor that can bind, but not be activated by BICDN and dynactin. What is the percentage of dynein motors that are not able to be activated, but co-localize with dynactin and BICDN in a single molecule assay on taxol-stabilized microtubules?

4) Despite the general high quality of the experimental data, the apparent number of experimental repeats is low. The authors state that the values shown represent 4 chambers per condition, with 15 microtubules analysed per chamber. But this doesn't clarify how many independent experiments were performed (should be a minimum of 3 independent experimental repeats, not just 3-4 chambers from a single experiment – perhaps they did this but it is unclear from their descriptions), and they do not specify the number of total events measured for each experiment.

5) Throughout the manuscript, protein levels are reported as molar ratios rather than absolute concentrations. However, a 1:10 dynein:Lis1 ratio can result in essentially all the dynein having Lis1 bound (if the concentrations are well above the Kd) or essentially no dynein having Lis1 bound (if the concentrations are well below the Kd). Therefore, it is essential to state the absolute final concentration of all species in the reactions in order to interpret them. This is especially important when comparing single-molecule and microtubule-gliding assays, in which the concentrations are presumably quite different.

6) Binding to the plus end is expressed per growth phase, rather than per unit time. This seems problematic for experiments +/- EB1, as EB1 substantially shortens the duration of the growth phase as shown in Figure 2—figure supplement 1?

7) Please include additional, or higher quality kymographs of dynein and BICD2 moving towards the minus end of dynamic microtubules (Figure 3).

8) Figure 6 appears confusing. It would help to break it into separate figures or change the presentation strategy.

9) It is well established that Clip170 interacts with LIS1 and the p150 subunit of dynactin (Lansbergen, et al. J. Cell. Biol., 2004) and that Clip170 plays an important role in dynein localization to microtubule plus ends (Roberts, et al. *eLife*, 2014; Duellberg, et al. Nat. Cell. Biol., 2014). The authors have purified and functional Clip170 in hand (Figure 1—figure supplement 1 and Figure 2—figure supplement 1). Have they tried to add it in their in vitro reconstitutions? In principle, it would be nice to know whether Clip170 affects the number and duration of dynein plus end tracking events in the presence of LIS1, dynactin, and EB1? If the authors have reasons not to include these experiments, they should at least include some discussion on this issue.

10)"Dynein complexes that track the plus ends of microtubules are unlikely to act directly as a source of motors for minus end-directed transport." This statement has the potential to mislead. While it may be true under these assay conditions (i.e. in the presence of a 10-fold molar excess of a constitutively active cargo adaptor fragment), the situation in the cell (where cargoes with bound adaptors are likely to be limiting) might be very different indeed. This should be acknowledged.

11) The discussion of the literature is selective, with the potential to mislead:

Nudel/NudE are not mentioned. However, these proteins are known to be critical for Lis1 functions in vivo, and relieve Lis1's inhibitory effect on mammalian dynein in vitro, and must be discussed.

The major of models of McKenney et al. (2010) and (2011) are not mentioned i.e. that Lis1 and NudE induce a persistent force-generating state in mammalian dynein, and that this is mutually exclusive with dynactin regulation.

Treatment of the yeast literature seems selective, with the inhibitory effects of Lis1/Pac1 in vitro played up but the substantial body of work elucidating Lis1/Pac1's role in vivo (consistent with a targeting/inhibitory function) played down. Notably, Lis1/Pac1 is absent from dynein's site of action in yeast, and evidence suggests it must be displaced in order to activate yeast dynein.

"It is unclear how [Lis1's] activities can account for the reduction in dynein-based processes that is generally observed when LIS1 function is inhibited in cells." This seems an oversimplification, as incorrect targeting/inhibitory regulation can perturb motor processes as much as loss of an activator.

"However, LIS1's specific function in plus end tracking of dynein was also not addressed in previous in vitro reconstitution assays." This is true for mammalian proteins, but Lis1/Pac1's role has been addressed in reconstitutions with yeast proteins.

"However, this [search-and-capture] mechanism has not been visualised directly." Plus-end capture and minus-end directed motility of melanophores has been visualized directly (Lomakin et al., 2009). Transfer of yeast dynein from the plus end to the cell cortex has been visualized directly (Markus et al., 2011).

There are other factors that could make plus end localization more complex in vivo as well. There are also additional mechanisms of dynein recruitment to the plus end and many other plus end binding proteins. Additional discussion of this complexity would strengthen the paper.

---

## [Author Response]

*Essential revisions:*

*1) The authors report a surprising ability of LIS1 to activate both the frequency and velocity of dynein-dependent runs. This observation appears to directly contrast with previous studies showing that LIS1 leads to dynein inhibition during gliding assays, and that the binding of LIS1 enhances the development of an ATPase insensitive state in which dynein remains bound to the microtubule. To the credit of the investigators, they repeat the gliding assay inhibition, which suggests that the difference is not due to differential activities of the purified dyneins used in the various labs. However, they provide no significant insights into the dramatic differences between their single molecule and gliding data, nor any insights into how their new observations can be reconciled with the many previous single molecule studies on LIS1 from the Reck-Peterson, Vallee, and Gross labs. Some explanation of the discrepancy between the single molecule and gliding assays should be provided. The authors should also provide much better controls and description for both the single molecule and gliding assays:*

We regret that we did not state explicitly that our results do not contradict the findings of previous studies in the literature, which were generated with very different experimental systems. For example, the work of McKenney, Gross, Vallee and colleagues assayed dynein behaviour in the absence of processivity activators and predominantly under load. Huang, Roberts, Reck-Peterson and colleagues studied yeast dynein, which has features distinct from mammalian dynein (dramatically reduced velocity and robust processive movement without other co-factors). We have now stated that our findings do not contradict those in the literature (subsection “Mammalian LIS1 enhances the frequency and velocity of minus end-directed dynein movements”, second paragraph). We believe that our improved coverage of the existing literature on LIS1’s regulation of dynein, which was requested by the reviewers, also helps clarify the relationship between our findings and those of other groups.

Most importantly, we have now performed two types of experiments that clarify the different effects of LIS1 in different assays. One set of experiments provides evidence that LIS1 stimulates the assembly of the dynein-dynactin-BICD2N complex (new Figure 5). This exciting finding offers an explanation for the ability of LIS1 to increase the frequency of minus end-directed transport in the presence of dynactin and BICD2N. Because dynactin and BICD2N were not included in previous single molecule studies of dynein, this effect of LIS1 would not have been evident. Our observation is, however, consistent with the previous finding by our group and the Zheng group that LIS1 can promote complex formation between dynein and dynactin in cells or cell extracts (Dix et al. 2013; Wang et al. 2013). Our data indicating that LIS1 is in fact sufficient to stimulate dynein-dynactin-BICD2N assembly are described in the second paragraph of the subsection “LIS1 increases the frequency and velocity of minus end-directed dynein movements”.

The mechanism by which mammalian LIS1 increases the velocity of dynein-dynactin-BICD2N complexes will require a separate in depth study, although our new data demonstrating that LIS1 can influence the interaction of dynein with dynactin and BICD2N suggests an exciting route for investigation. The second set of new experiments that address the effects of LIS1 in different types of assays is described in response to point 1A. Collectively, our results are consistent with LIS1 exerting different effects on the activity of mammalian dynein in the presence and absence of dynactin and BICD2N.

*A) The authors should investigate whether Lis1 has an effect on human dynein microtubule interactions at the single molecule level, in the absence of other regulators (landing rate, dwell time, balance of static vs. diffusive events). The focus of this study is on more complex reconstitutions, but these data are fundamental and it seems very likely the authors already have them. Figure 2 compares dynein behaviour +/- Lis1 and dynactin, but the crucial +Lis1 kymograph seems to be missing from panel A.*

We thank the reviewers for this excellent suggestion. We have performed a set of new experiments investigating the effects of LIS1 on the single molecule interactions of human dynein complexes with microtubules (new Figure 3 and Figure 3—figure supplement 1). These experiments use stabilised microtubules, which facilitates quantification of dynein binding events.

The published in vitro studies of the influence of LIS1 on interactions of yeast dynein with microtubules seemingly used a large molar excess of LIS1. Our new experiments therefore not only used the relative concentrations of LIS1 and dynein used in the rest of our single molecule experiments, but also a 15-fold further excess of LIS1 to dynein. There was no significant influence of the lower concentration of LIS1 on the interactions of dynein with the microtubule lattice. This finding is consistent with our observations with dynamic microtubules. However, the higher concentration of LIS1 significantly increased the landing rate of dynein on microtubules without a detectable effect on the dwell time of the motor. (Diffusive events of dynein were very rare in the presence and absence of LIS1 and therefore analysis of static vs diffusive behaviour of the motor complex was not appropriate.)

Our finding that LIS1 can increase the landing rate of dynein is compatible with previous observations that LIS1 increases the association of other mammalian dyneins with microtubules in co-sedimentation assays (subsection “LIS1 stimulates plus end tracking of dynein complexes on growing microtubules in the presence of EB1 and dynactin”, last paragraph). We point out that our results with human proteins differ from those of the study with yeast proteins (Huang, Roberts et al. 2012), which found that LIS1 increases the dwell time of monomeric dynein on microtubules. This finding suggests that LIS1 can exert different effects on dynein’s interactions with microtubules in yeast and mammals, a potentially significant finding when seeking to reconcile results from these systems.

Although further mechanistic studies need to be performed in the future, we believe our new experiments also offer a possible explanation for the effects of LIS1 on mammalian dynein in different assays. We speculate that the ability of high relative concentrations of LIS1 to increase the association of human dynein with microtubules contributes to the inhibitory effect of LIS1 on microtubule translocation (subsection “Mammalian LIS1 enhances the frequency and velocity of minus end-directed dynein movements”, second paragraph). At lower relative concentrations of LIS1, such as those used in our other single molecule assays, there is no effect on the interaction of dynein with microtubules unless co-factors, such as plus end-associated dynactin complexes, are also present. We also point out that the ability of LIS1 to prolong the interaction of mammalian dynein with microtubules under load (McKenney et al. 2010) may also contribute to the inhibition of microtubule gliding by LIS1.

We have now added representative kymographs of dynamic microtubules incubated with human dynein alone and in the presence of LIS1 to Figure 2, as requested. We observed no effect of LIS1 on the interactions of dynein with microtubules from inspection of many kymographs. We have also modified Figure 2 to include our failure to observe end tracking of dynein in these conditions.

*B) A major experimental concern relates to the microtubule gliding assay (Figure 4), which the authors use to report that their Lis1 preparation reduces the velocity of microtubule gliding driven by human dynein. Apparently, dynein was non-specifically adsorbed to the flow chamber either alone or in the presence of 6 μm or 60 μm Lis1. Unbound proteins were then washed out. This approach raises two substantive concerns.*

*First, did the presence of Lis1 (up to ~2.7 mg/ml) reduce the amount of dynein that non-specifically adsorbed to the surface? Lis1 is a sticky protein, so this seems plausible, especially at the 200-fold Lis1 molar excess. The presence of microtubules on the surface does not rule this out, as microtubules adsorb to many surfaces, including those lacking motors. Either the consistency of dynein surface density in the different conditions needs to be confirmed (e.g. using fluorescent dynein) or, to avoid this danger entirely, the authors could specifically immobilize dynein, then subsequently add the desired concentration of Lis1.*

We mixed dynein with LIS1 prior to washing off unbound motor from the glass surface as we thought that this was the best way to combine the proteins in a defined molar ratio. Controlling molar ratios is not practical when LIS1 is added to chambers with prebound dynein because the amount of motor complex bound to the surface cannot be readily quantified. Nonetheless, the reviewers’ comments made us appreciate that additional controls are needed to support our interpretations of the gliding assays.

In the assay conditions employed, microtubules do not associate with the glass surface when dynein is omitted. We now cite a recent publication from our group that includes this control (Hoang et al. 2017). This finding suggests that LIS1 did not inhibit gliding by causing strong dissociation of the motor complex from the glass, as this would inhibit microtubule association. As suggested by the reviewers, we have now investigated directly the effects of LIS1 on the association of dynein with the surface using the fluorophore on the heavy chain. We show that in conditions in which LIS1 strongly inhibits microtubule gliding there is no reduction in the amount of motor complex on the glass surface (Figure 6—figure supplement 3). In these experiments LIS1 was added to dynein that was already immobilised on the surface to rule out competition for initial binding to the glass, as recommended by the reviewers.

We also performed an additional control showing that LIS1 alone does not cause microtubules to adhere to the glass surface in our assay buffer (Video 2). This finding, which was also reported in Yamada et al. (2008) in their study with porcine dynein, is valuable as it indicates that LIS1 does not indirectly inhibit microtubule gliding by dynein by providing a strong independent attachment between microtubules and the glass surface. These results are also consistent with those of Vallee, Gross and colleagues (McKenney et al. 2010), who used other methods to show that LIS1 does not interact with microtubules. We therefore show with two different experimental configurations that in the presence of LIS1 microtubules associate with the glass surface through human dynein but are translocated inefficiently. The new experiments have led to several modifications to this section of the Results (subsection “LIS1 increases the frequency and velocity of minus end-directed dynein movements”, last paragraph).

One of the reviewers’ comments led us to check the concentrations of dynein and LIS1 in our previous experiments. There was an unfortunate miscalculation when compiling the previous Materials and methods; the concentration of dynein in the solution incubated with the glass surface was 30 nM, and the concentration of LIS1 was 0.6 or 6 μM. This error has been corrected and we have confirmed that protein concentrations in all other assays are stated correctly.

*Second, washing the chamber in a solution lacking Lis1 means that the dynein-Lis1 binding reaction will not have been at the desired equilibrium during the experiment: Lis1 will have been gradually dissociating in the time post wash out, and thus the fraction of dynein molecules with Lis1 bound will have depended critically on the time that elapsed between the washout and the start of the data acquisition. Was this elapsed time kept constant? It would be much safer to include Lis1 at the desired concentration in the final assay mixture.*

We kept the time between addition of proteins and imaging constant in our previous image series and observed no diminution over time of the inhibitory effect of LIS1 on microtubule gliding. This was also the case in our new experimental series. We illustrate this point in Figure 10 using data from the new gliding assays.

Author response image 1.Comparison of microtubule gliding velocity during the first and second 5 minutes after the beginning of image acquisition.Magenta circles: values for individual microtubules. Error bars: S.D.**DOI:**
http://dx.doi.org/10.7554/eLife.21768.026

Thus, there is sufficient LIS1 associated with dynein to exert a strong inhibitory effect for the duration of the gliding assay. Including LIS1 with the microtubules in the final assay mixture is therefore not necessary (it is also not practical due to the requisite dilution factor for LIS1). We have now made it clear in the Materials and methods that the time between protein addition and image acquisition was kept constant and that there was no reduction in LIS1’s inhibitory effect over the time course of the experiment.

*2) The experiments with the dynein tail construct are somewhat confusing. Is the tail alone sufficient to track growing and shrinking plus ends? What is the minimum set of proteins required to allow the tail to track growing and shrinking ends? Can this tracking be seen with purified dynein, or is it limited to recombinant forms? Does Lis1 influence the frequency/duration of tracking events for the dynein tail, or is the motor domain required? (Alternatively, mutations that prevent Lis1 binding at the AAA4 site are known, and could be tested in the tracking assay.)*

*The authors propose the model that the different effects of Lis1 may arise from different binding sites, invoking interactions reported between Lis1 and various fragments of dynein and dynactin. However, interactions reported from the cited yeast 2-hybrid and pull-down experiments have not all been reproduced with intact proteins, so it is necessary to exercise caution before setting the field off down this path. Importantly, it seems that the authors may already have the data to test this model in their end-tracking experiments with the dynein tail construct. i.e. if Lis1 influences the frequency/duration of tracking events for the dynein tail, this would provide strong evidence for a functional binding site outside the characterised one in the dynein motor domain.*

The observation with the tail was a minor part of the study and was not emphasised in the Discussion because of several open questions, which were also picked up on by the reviewers. We could not previously address the minimal requirements for tracking of the dynein tail as the lack of a fluorescently labelled tail construct meant that the behaviour of the tail had to be visualised by fluorescent BICD2N, which only associates with the tail in the presence of dynactin. In this assay configuration it was also not possible to determine if LIS1 promotes end tracking of the isolated dynein tail; altered end tracking behaviour of labelled BICD2N could also result from an affect of LIS1 on the association of the dynein tail with BICD2N and dynactin. We also could not compare the efficiency of tracking by the dynein tail versus the full complex using labelled BICD2N as the results could be confounded by any differences in the efficiency of dynactin-dependent binding of BICD2N to the tail compared to the full dynein complex. We agree that the experiments with the dynein tail construct were therefore confusing and unsatisfactory. We were unable to convincingly address the questions raised above in the time available for revision, and have therefore decided to remove the two panels in the old Figure 6 that refer to the tail. We believe this makes this section of the manuscript much clearer.

We have retained some discussion in the Perspective about the possibility of LIS1 having more than one binding site on the dynein-dynactin complex as we think this is an attractive hypothesis for how LIS1 exerts context-dependent (and concentration-dependent) effects. We have now stated that the earlier evidence for LIS1 interactions with the tail and p50 was from yeast two hybrid and pulldown methods, which should allow the reader to exercise appropriate caution when reading this section. We now make it clear that this section is speculation and not a proposed model. Incidentally, we would not have been able to address the reviewers’ question about the tracking behaviour of the native dynein tail, as we can only produce the tail in a recombinant form.

*3) It is possible that dynein-dynactin-BICDN complexes that track the plus end contain a "dead" motor (Figure 3), or a dynein motor that can bind, but not be activated by BICDN and dynactin. What is the percentage of dynein motors that are not able to be activated, but co-localize with dynactin and BICDN in a single molecule assay on taxol-stabilized microtubules?*

We had also considered this possibility but not looked into it further. We have now followed the reviewers’ helpful suggestion and quantified the percentage of all static dynein motors on the lattice that co-localise with BICD2N (and therefore dynactin). Quantification was performed using our previous kymographs of dynamic microtubules, so that the results could be compared directly to our analysis of BICD2N and dynein co-localisation on plus end tracking complexes in the presence of dynactin. Approximately 60% of static dynein complexes on the lattice had a BICD2N signal (Figure 4—figure supplement 3), which is not dissimilar to the percentage of plus end tracking dyneins that co-localised with BICD2N (Figure 4). Thus, dynein-dynactin-BICD2N complexes that track the plus end in the presence of EB1 could contain a dynein motor that cannot be activated by dynactin and BICD2N alone. This point is now made in the last paragraph of the subsection “End tracking dynein complexes are rarely a source of motor for minus end-directed movements”. The revised manuscript has much less discussion of potential mechanisms underlying initiation of processive movement from plus end tracking events as we agree with the reviewers that it will be important to model this process in vitro with additional factors, including native cargo, cargo adaptors and NudE (subsection “Insights into plus end tracking of dynein complexes and the relationship to cargo transport”, last paragraph and subsection “Perspective”). We have instead emphasised the functions of LIS1 in regulating dynein activity, including the results of new experiments and analyses suggested by the reviewers.

*4) Despite the general high quality of the experimental data, the apparent number of experimental repeats is low. The authors state that the values shown represent 4 chambers per condition, with 15 microtubules analysed per chamber. But this doesn't clarify how many independent experiments were performed (should be a minimum of 3 independent experimental repeats, not just 3-4 chambers from a single experiment – perhaps they did this but it is unclear from their descriptions), and they do not specify the number of total events measured for each experiment.*

We apologise for not including this information in the first submission. Each chamber is derived from a unique assembly mix, which we operationally define as an independent experiment. For each chamber, movies of three different regions were acquired. Different experimental conditions in the same series where typically performed in an interleaved manner on the same day. The data set for each experimental condition comprises chambers from at least two different days, and all the data were analysed carefully to check that there is no significant day-to-day variability between results. This information is now included in the Materials and methods. Information on the number of events analysed per condition is now included in each figure legend. A very large number of events was analysed throughout the study, and all of this was done manually. Consistent effects in the presence and absence of LIS1 were also seen in independent experimental series (e.g. ± BICD2N, stabilised versus dynamic microtubules).

*5) Throughout the manuscript, protein levels are reported as molar ratios rather than absolute concentrations. However, a 1:10 dynein:Lis1 ratio can result in essentially all the dynein having Lis1 bound (if the concentrations are well above the Kd) or essentially no dynein having Lis1 bound (if the concentrations are well below the Kd). Therefore, it is essential to state the absolute final concentration of all species in the reactions in order to interpret them. This is especially important when comparing single-molecule and microtubule-gliding assays, in which the concentrations are presumably quite different.*

We have now added information to the Materials and methods and figure legends on the concentration of dynein in the assembly mixes.

*6) Binding to the plus end is expressed per growth phase, rather than per unit time. This seems problematic for experiments +/- EB1, as EB1 substantially shortens the duration of the growth phase as shown in Figure 2—figure supplement 1?*

We confirmed that the duration of growth phases is not significantly different between conditions in key experiments comparing the frequency of dynein binding events per growth phase (Figure 2—figure supplement 2; Figure 4—figure supplement 1; Figure 4—figure supplement 2). We did not perform this analysis for some of the conditions in Figure 2 in which end tracking was essentially not observed through the entire field in several movies. However, the lack of tracking events in the presence or absence of EB1 when dynactin is omitted indicates that this result is independent from effects of EB1 on the duration of growth phases.

*7) Please include additional, or higher quality kymographs of dynein and BICD2 moving towards the minus end of dynamic microtubules (Figure 3).*

We have now included an additional kymograph of dynein and BICD2N complexes moving towards the minus end of dynamic microtubules (Figure 4—figure supplement 3). Please note that the quality of these kymographs is not as high as of those obtained from single or dual colour imaging as the overall acquisition rate is reduced by the capture of another channel.

*8) Figure 6 appears confusing. It would help to break it into separate figures or change the presentation strategy.*

We now see that the inclusion of the shrinking end analysis with the labelled BICD2N and the dynein tail in the same figure with the results of the experiments with labelled full-length dynein was confusing. This is no longer an issue as the tail experiments have been removed from the manuscript for the reasons described above. To further simplify the figure, we have moved the data for experiments without dynactin to the supplement (Figure 8—figure supplement 1).

*9) It is well established that Clip170 interacts with LIS1 and the p150 subunit of dynactin (Lansbergen, et al. J. Cell. Biol., 2004) and that Clip170 plays an important role in dynein localization to microtubule plus ends (Roberts, et al. eLife, 2014; Duellberg, et al. Nat. Cell. Biol., 2014). The authors have purified and functional Clip170 in hand (Figure 1—figure supplement 1 and Figure 2—figure supplement 1). Have they tried to add it in their* in vitro *reconstitutions? In principle, it would be nice to know whether Clip170 affects the number and duration of dynein plus end tracking events in the presence of LIS1, dynactin, and EB1? If the authors have reasons not to include these experiments, they should at least include some discussion on this issue.*

We had performed experiments with CLIP-170 but had not analysed the experiments as we planned to focus on the effects of this protein in a future study. The reviewer’s comment made us appreciate that readers would be interested in the effects of CLIP-170 in the current study. We have now analysed the experiments and added the data to the manuscript. We show that, in the presence of LIS1, CLIP-170 has no detectable effect on the ability of EB1 to target the initiation of processive transport events to dynein-dynactin-BICD2N complexes to growing plus ends, or on the frequency and duration of plus end tracking events (Figure 4—figure supplement 2). This finding is compatible with the conclusion of Duellberg et al. (2014) that CLIP-170 only becomes important for plus end tracking of p150 in the presence of competing EB1-binding peptides (subsection “EB1 can direct dynein-dynactin-BICD2N transport initiation to growing plus ends but this is not influenced by LIS1 or CLIP-170”, last paragraph).

*10)"Dynein complexes that track the plus ends of microtubules are unlikely to act directly as a source of motors for minus end-directed transport." This statement has the potential to mislead. While it may be true under these assay conditions (i.e. in the presence of a 10-fold molar excess of a constitutively active cargo adaptor fragment), the situation in the cell (where cargoes with bound adaptors are likely to be limiting) might be very different indeed. This should be acknowledged.*

This is another very good point. We now make it explicit that we are referring to the findings in our assay conditions. We agree with the reviewer that several other factors could influence the initiation of processive movement from the plus end in vivo. We discuss the possibility that the cargo or microtubule modifications could be involved in a physiological setting (subsection “Insights into plus end tracking of dynein complexes and the relationship to cargo transport”, last paragraph). In response to the reviewers’ comments we now also include experiments assaying the effects of CLIP-170 (see above), and mention the importance of assessing the effects of NudE proteins in the future (subsection “Perspective”).

*11) The discussion of the literature is selective, with the potential to mislead:*

We thank the reviewers for these suggestions for improving the discussion of the literature. We have made several changes.

*Nudel/NudE are not mentioned. However, these proteins are known to be critical for Lis1 functions* in vivo*, and relieve Lis1's inhibitory effect on mammalian dynein* in vitro*, and must be discussed.*

We now mention prominently in the Perspective section that addressing the role of NudE/NudEL is a priority for future experiments (see above).

*The major of models of McKenney et al. (2010) and (2011) are not mentioned i.e. that Lis1 and NudE induce a persistent force-generating state in mammalian dynein, and that this is mutually exclusive with dynactin regulation.*

We had cited the McKenney et al. (2010) paper but realise that we should have introduced the model explicitly. We now do so in the Introduction (seventh paragraph) and Discussion (subsection “Mammalian LIS1 enhances the frequency and velocity of minus end-directed dynein movements”, second paragraph). The 2011 McKenney et al. paper focuses on the competition between NudE and dynactin. In the absence of NudE experiments in our study, we had not cited this paper in our first submission. The one experiment with LIS1 in the 2011 study shows that it promotes the ability of the CC1 domain of p150 to release dynein from NudE, which suggests a complex set of interactions between these factors. The revised manuscript introduces the need to study NudE/NudEL in our reconstitution assays in the future and we now cite the McKenney et al. 2011 paper in this context.

*Treatment of the yeast literature seems selective, with the inhibitory effects of Lis1/Pac1 in vitro played up but the substantial body of work elucidating Lis1/Pac1's role in vivo (consistent with a targeting/inhibitory function) played down. Notably, Lis1/Pac1 is absent from dynein's site of action in yeast, and evidence suggests it must be displaced in order to activate yeast dynein.*

We now emphasise in the Introduction that there is substantial evidence for a targeting/inhibitory function of Lis1/Pac1 from different studies. This section includes citations to papers that elucidate the effect of Lis1/Pac1 on dynein in yeast. These citations are also used in the Results and Discussion. We expand on the findings of the Roberts et al. study from 2014 to state that the LIS1 and CLIP170 orthologues from yeast can couple dynein to a plus end-directed motor, Kip2.

*"It is unclear how [Lis1's] activities can account for the reduction in dynein-based processes that is generally observed when LIS1 function is inhibited in cells." This seems an oversimplification, as incorrect targeting/inhibitory regulation can perturb motor processes as much as loss of an activator.*

We include more discussion of other models for how LIS1 can promote dynein cargo transport in vivo (Introduction, seventh paragraph). We have been careful not to give the impression that our results contradict these models. However, they do offer an additional explanation for how LIS1 promotes cargo transport in the presence of processivity activators, which we find exciting. We believe our new evidence that LIS1 can promote the assembly of the dynein-dynactin-BICD2N complex strengthens this aspect of the manuscript significantly.

*"However, LIS1's specific function in plus end tracking of dynein was also not addressed in previous* in vitro *reconstitution assays." This is true for mammalian proteins, but Lis1/Pac1's role has been addressed in reconstitutions with yeast proteins.*

We now refer specifically to mammalian proteins in this section. We also mention in the Discussion specific findings of the reconstitution studies with yeast proteins.

"However, this [search-and-capture] mechanism has not been visualised directly." Plus-end capture and minus-end directed motility of melanophores has been visualized directly (Lomakin et al., 2009). Transfer of yeast dynein from the plus end to the cell cortex has been visualized directly (Markus et al., 2011).

*There are other factors that could make plus end localization more complex in vivo as well. There are also additional mechanisms of dynein recruitment to the plus end and many other plus end binding proteins. Additional discussion of this complexity would strengthen the paper.*

The restructuring of the Introduction to give more emphasis to existing models of how LIS1 regulates dynein (in response to the points above) has resulted in the section on visualisation of search-and-capture by plus end-associated dynein in vivo being removed. We now make it more explicit in the Discussion and the Perspective that additional factors could be involved in regulating transport initiation and plus end localisation in vivo. The Lomakin et al. and Markus et al. studies are cited elsewhere in our manuscript.